# Nano-HTDMA for investigating hygroscopic properties of sub-10 nm aerosol nanoparticles

Ting Lei[1,2], Nan Ma[4,1,3], Juan Hong[4,1], Thomas Tuch[3], Xin Wang[2], Zhibin Wang[5], Mira Pöhlker[2], Maofa Ge[6], Weigang Wang[6], Eugene Mikhailov[7], Thorsten Hoffmann[8], Ulrich Pöschl[2], Hang Su[2], Alfred Wiedensohler[3], Yafang Cheng[1]

[1]Minerva Research Group, Max Planck Institute for Chemistry, 55128 Mainz, Germany

[2]Multiphase Chemistry Department, Max Planck Institute for Chemistry, 55128 Mainz, Germany

[3]Leibniz Institute for Tropospheric Research, 04318 Leipzig, Germany

[4]Institute for Environmental and Climate Research, Jinan University, 511443 Guangzhou, China

[5]Reserch Center for Air Pollution and Health, College of Environmental and Resource Science, Zhejiang University, Hangzhou, 310058, China

[6]Beijing National Laboratory for Molecular Sciences (BNLMS), Institute of Chemistry, Chinese Academy of Sciences, Beijing, 100190, P. R. China

[7]St. Petersburg State University, 7/9 Universitetskaya nab., St. Petersburg, 199034, Russia

[8]Institute of Inorganic Chemistry and Analytical Chemistry, Johannes Gutenberg University Mainz, Mainz, Germany

*Correspondence to*: Yafang Cheng (yafang.cheng@mpic.de) and Juan Hong (juanhong0108@jnu.edu.cn)

**Abstract.** Interactions between water and nanoparticles are relevant for atmospheric multiphase processes, physical chemistry, and materials science. Current knowledge of the hygroscopic and related physico-chemical properties of nanoparticles, however, is restricted by limitations of the available measurement techniques. Here, we present the design and performance of a nano-hygroscopicity tandem differential mobility analyzer (nano-HTDMA) apparatus that enables high accuracy and precision in hygroscopic growth measurements of aerosol nanoparticles with diameters less than 10 nm. Detailed methods of calibration and validation are provided. Besides maintaining accurate and stable sheath/aerosol flow rates (±1%), high accuracy of DMA voltage (±0.1%) in the range of ~0-50 V is crucial to achieve accurate sizing and small sizing offsets between the two DMAs (<1.4%). To maintain a stable relative humidity (RH), the

humidification system and the second DMA are placed in a well-insulated and air conditioner housing
(±0.1K). We also tested and discussed different ways of preventing pre-deliquescence in the second DMA.
Our measurement results for ammonium sulfate nanoparticles are in good agreement with Biskos et al.
(2006b), with no significant size-effect on the deliquescence and efflorescence relative humidity (DRH,
ERH) at diameters down to 6 nm. For sodium sulfate nanoparticles, however, we find a pronounced size-
dependence of DRH and ERH between 20 and 6 nm nanoparticles.

## 35  1 Introduction

The climatic effects of aerosol nanoparticles have attracted increasing interest in recent years (Wang et al.,
2016; Andreae et al., 2018; Fan et al., 2018). Interactions between water and nanoparticles are relevant for
atmospheric multiphase processes, physical chemistry, and materials science (Zheng et al., 2015; Cheng et
al., 2015, 2016). Aerosol nanoparticles in the atmosphere are mostly originating from new particle
formation, and a fraction of these nanoparticles could potentially grow into sizes to efficiently act as cloud
condensation nuclei and thus to change the contributions of aerosol nanoparticles to climate forcing
(Lihavainen 2003; Wiedensohler et al., 2009; Sihto et al., 2011; Kirkby et al., 2011; Keskinen et al., 2013;
Dunne et al., 2016; Kim et al., 2016). These processes strongly depend on the chemical composition and
physico-chemical properties of these nanoparticles (Köhler, 1936; Su et al., 2010; Wang et al., 2015; Cheng
et al., 2015). One of the most important physico-chemical properties of nanoparticles is their hygroscopic
behavior that describes their ability to take up water, and it can differ significantly from that of larger
particles (Hämeri et al., 2000, 2001; Gao et al., 2006; Biskos et al., 2006a, b, 2007; Cheng et al., 2015).
To understand and predict hygroscopic properties of nanoparticles, current thermodynamic models mostly
rely on the concentration-dependent thermodynamic properties (such as water activity and interfacial
energy) derived from the measurements of large aerosol particles or even bulk samples (Tang and
Munkelwitz, 1994; Tang 1996; Pruppacher and Klett, 1997; Clegg et al., 1998). They are thus difficult or

impossible to be applied to describe the hygroscopic behavior of sub-10 nm nanoparticles which can be often supersaturated in concentration compared to bulk solutions (Cheng et al., 2015). Furthermore, the nanosize effect on these properties may also need to be considered (Cheng et al., 2015). The lack of such data hinders the understanding and the accurate simulation of the interaction of water vapor and atmospheric nanoparticles. In addition, by knowing the hygroscopicity of newly formed nanoparticle, one can infer the involving chemical species (e.g., organic ratio) in particle formation and initial growth (Wang et al., 2010), which is otherwise difficult and highly challenging to measure directly (Wang et al., 2010; Ehn et al., 2014). Hence, to measure the hygroscopicity of nanoparticles is essential to improve our understandings of aerosol formation, transformation, and their climate effects.

Different techniques have been employed to characterize the hygroscopic properties of aerosol particles in different sizes (Fig. S1) (Tang et al., 2019), such as Fourier transform infrared spectrometer (FT-IR) (Zhao et al., 2006), Raman spectroscopy (Dong et al, 2009), electrodynamic balance (EDB) (Chan and Chan, 2003, 2005; Chan et al., 2008), optical tweezers (Reid et al., 2011; Rickards et al., 2013), hygroscopicity tandem differential mobility analyzer (HTDMA) (e.g., Rader and McMurry, 1986; Mikhailov et al., 2004; 2008; 2009; Biskos et al., 2006a, b, 2007; Cheng et al., 2008, 2009; Eichler et al., 2008; Stock et al., 2011; Hong et al., 2014, 2015; Lei et al., 2014; 2018; Mikhailov and Vlasenko, 2019), and atomic force microscopy (AFM) (Estillore et al., 2017). Using these techniques, most of the early lab studies focused on the hygroscopic behavior of particles in accumulation modes and super-micron size range, including deliquescence, efflorescence of pure components and the effect of organics on the change or suppression of deliquescence and efflorescence of these inorganic components in mixtures.

For nanoparticles with diameters down to sub-10 nm, there are, however, only very few studies that have attempted to investigate their interactions with water molecules, which mainly utilized the setup with humidified tandem DMAs (Hämeri et al., 2000, 2001; Sakurai et al., 2005; Biskos et al., 2006a, b, 2007; Giamarelou et al., 2018). In Table S1, we summarized the measured DRH and ERH of ammonium sulfate

nanoparticles in the size range from 6 to 100 nm using HTDMAs. In these studies, the results of the observed DRH and ERH and prompt or non-prompt phase transitions of ammonium sulfate nanoparticles, however, do not show universal agreement. The technical challenges in HTDMA measurements, especially in the sub-10 nm size range, mainly lie on: (1) accurate sizing and small sizing offset of the two DMAs, (2) highly stable measurement conditions in the whole system. Large sizing offset between the two DMAs may lead to significant error in the measured growth factor based on error propagation (Mochida and Kawamura, 2004). Massling et al. (2011) and Zhang et al. (2016) suggested that to achieve good hygroscopic growth factor of nanoparticles, the sizing offset of the two DMAs should be within ±2-3%, which is however very difficult to maintain for the sub-10 nm size range. To accurately measure phase transition (e.g., DRH and ERH), a highly stable measurement condition is essential, especially maintaining a small temperature perturbation in the humidification system and inside the second DMA to prevent pre-deliquescence. For example, a 0.8 K fluctuation of the experimental temperature during the measurement can result in a 4% difference in RH (0-90%) inside the humidified DMA (Hämeri et al., 2000), leading to an inaccurate determination of the phase transition. Another problem is the prompt versus non-prompt phase transition. Although effects of impurities on the phase transition of aerosol nanoparticles (Biskos et al., 2006a; Russell and Ming, 2002) may be one possible reason of the previously observed non-prompt phase transitions (e.g., Hämeri et al., 2000), the apparent non-prompt phase transition of aerosol nanoparticles has been thought to be mainly due to the inhomogeneity of RH and temperature in the humidified DMA during measurements (Biskos et al., 2006b; Bezantakos et al., 2016). Moreover, the hygroscopic measurements are in general difficult for nanoparticles with diameters below 20 nm due to high diffusion losses of nanoparticles (Seinfeld and Pandis, 2006).

In this study, we present the design of a nano-HTDMA setup that enables high accuracy and precision in hygroscopic growth measurements of aerosol nanoparticles with diameters less than 10 nm. Detailed methods of calibration and validation are provided. We discuss in detail how to maintain the good performance of the system by minimizing uncertainties associated with the stability and accuracy of RH,

temperature, voltage for nanoparticle classification, and sheath and aerosol flows in the DMA systems. We
then apply the nano-HTDMA system to study the size dependence of the deliquescence and the
efflorescence of aerosol nanoparticles of two specific inorganic compounds (e.g., ammonium sulfate and
sodium sulfate) for sizes down to 6 nm.

## 106    2. Methods

### 107    2.1 Nano-HTDMA system

We designed a nano-HTDMA system to measure the aerosol nanoparticle hygroscopic growth factor ($g_f$),
especially aiming for accurate measurement of phase transition and hygroscopic growth factor for
nanoparticles in the sub-10 nm size range. Here, $g_f$ is defined as the ratio of mobility diameters of
nanoparticles after humidification ($D_m(RH)$) to that in the dry condition ($D_m(< 10\% \text{ RH})$) (see SI. S1. Eq.
(S1)). As presented in Fig. 1, the nano-HTDMA composes three main components, including two nano-
differential mobility analyzers (nano-DMA, TROPOS Model Vienna-type short DMA; Birmili et al., 1997),
an ultrafine condensation particle counter (CPC, TSI Model 3776), and a humidification system. Table 1
shows the technical specification, where the DMA system, humidification system, and temperature system
of the three HTDMAs setup are compared among the systems of Biskos et al. (2006b), Hämeri et al. (2000)
and this study.
In our setup (Fig. 1), the first nano-DMA (nano-DMA1) is used to produce quasi-monodisperse
nanoparticles at a desired dry diameter. The flow rate of the closed-loop sheath flow in the nano-DMA1 is
maintained at 10 l/min. The ratio of sheath flow to aerosol flow is 10:1.5. The sheath flow is dried to RH
below 10% by two custom-built Nafion dryers (TROPOS Model ND.070) in parallel. The quasi-
monodisperse nanoparticles produced by nano-DMA1 then enter the humidification system, which can be
set to deliquescence mode (from low RH to high RH for measuring deliquescence) or efflorescence mode

(from high RH to low RH for measuring efflorescence). In the deliquescence mode, dry nanoparticles are humidified by a Nafion humidifier (NH-1, TROPOS Model ND.070, Length 60 cm) to a target RH. In the efflorescence mode, nanoparticles are first exposed to a high RH condition (~97% RH) in a Nafion humidifier (NH-2, Perma Pure Model MH-110, Length 30 cm) and then dried to a target RH through NH-1. The humid flow in the outer tube of NH-1 is a mixture of high-humidity air produced with a custom-built Gore-Tex humidifier and heater (GTHH: TROPOS Model, Inner Radius 1.5 cm & Length 30 cm) and dry air in variable proportions. To have precise control of the aerosol RH, the flow rates of the humid and dry air are adjusted with a proportional-integral-derivative (PID) system, including two mass flow controllers (MFC: MKS Model MF1) and an RH sensor (Vaisala Model HMT330) downstream of NH-1.

The residence time is ~5.4 s in the NH-1 for both the deliquescence and the efflorescence modes. Many groups have reported that the residence time of a few seconds is sufficient to reach equilibrium for measuring hygroscopic growth or shrink of inorganic salt particles, e.g., ammonium sulfate and chloride sodium (Chan and Chan, 2005; Duplissy et al., 2009; Lei et al., 2014, 2018; Giamarelou et al., 2018). More specifically, Kerminen (1997) estimated the time for reaching the water equilibrium to be between $8 \times 10^{-6}$ s and 0.005 s for 100 nm nanoparticles at 90% RH at 25°C with accommodation coefficients from 0.001 to 1, respectively. In our study, we measured the inorganic aerosol nanoparticles with diameters from ~100 nm down to 6 nm, thus the equilibrium time should be even shorter as nanoparticle size decreases (Table. S2). In NH-2, the residence time is ~0.07 s for the deliquescence of inorganic aerosol nanoparticles at very high RH condition (~97% RH), which is much longer than the time estimated for phase transition by Duplissy et al. (2009) (in the order of a few milliseconds) and Raoux et al. (2007) (in the order of a few nanoseconds). In addition, we have tested a longer NH-2 (Perma Pure Model MH-110, Length 121 cm) in the efflorescence mode, and no significant difference in measured growth factors is found, indicating that the residence time in NH-1 and NH-2 should be sufficient.

The number size distribution of the humidified nanoparticles is measured with a combination of the second
nano-DMA (nano-DMA2) and the ultrafine CPC. Similar to Biskos et al. (2016b), a multiple Nafion
humidifier (NH-3, Perma Pure Model PD-100) is used in our nano-HTDMA system to rapidly adjust the
RH of the sheath flow of nano-DMA2. The sheath flow is fed into the outer tube of NH-3 to minimize its
pressure drop. The RH of humid flow in the inner tube of NH-3 is controlled with a similar PID system as
that for NH-1. An RH sensor (Vaisala Model HMT330) downstream of NH-3 is used to provide feedback
to the PID system. In our nano-HTDMA system, a dew point mirror (DPM: EDGE TECH Model MIRROR-
99) is placed in the excess flow line to measure the RH and temperature of excess flow of the nano-DMA2.
During the operation, the difference between sheath flow RH and aerosol flow RH has been maintained
within ±1% (see more details in Section 2.2).
The sheath flow is maintained to the set flow rate with a PID-controlled recirculation blower (RB:
AMETEK Series MINISPIRAL). Prior to every size scan, the sheath flow rate of nano-DMA2 is adjusted
by the PID system according to the measurement of a mass flow meter (MFM: TSI Series 4000) in the
sheath flow line. In order to minimize the pressure drop along the recirculating sheath flow loop, a low flow
resistance MFM and hydrophobic filter (HF: Whatman Model 6702-3600) are used. A heat exchanger (HE,
Ebmpapst Model 4414FM) is installed downstream of the RB to minimize the temperature perturbation in
the sheath flow by the heat generated in the RB.
As aforementioned, temperature non-uniformity is the main contributor to the fluctuation of RH within
humidified DMA. Temperature difference within nano-DMA2 is unavoidable mainly due to temperature
difference between the inner electrode and the rest of nano-DMA2 parts and/or the temperature difference
between aerosol and sheath flow (Duplissy et al., 2009; Bezantakos et al., 2016). As shown in Fig. 1, to
investigate and monitor the temperature difference within nano-DMA2 during measurements, a
temperature sensor (THERMO ELECTRON Model Pt100) is placed at the inlet of the sheath flow and the
temperature of sheath excess flow is monitored by the DPM. Note that, a DPM should be installed as close
as possible to the nano-DMA2 in the excess flow, which better represents the conditions inside the nano-
DMA2, such as temperature and RH (Wiedensohler et al., 2012). In addition, the temperature of aerosol
flow is monitored at the inlet of the aerosol flow of nano-DMA2.
Moreover, to maintain a stable environment that required for the growth factor measurements, nano-DMA2
with its sheath flow humidification system is placed in a well-insulated housing chamber (marked with
yellow dashed lines in Fig. 1). An air conditioner (Telemeter Electro Model TEK-1004-RR-24-IP55) is
installed inside the housing to maintain a constant temperature ($292.15\pm0.1$ K), which is set to be ~1 K
lower than the constant laboratory temperature (293 K) in order to achieve high RH (~90%) inside nano-
DMA2.

## 2.2 Calibration of nano-HTDMA

The purpose of this study is to design and build a nano-HTDMA system that is able to measure the
hygroscopic properties of nanoparticles, especially in the sub-10 nm size range. A small perturbation in the
measurement conditions may lead to large biases in the results. Hence, to provide high quality
hygroscopicity measurements of nanoparticles, systematic calibration of the nano-HTDMA should be
conducted regularly to ensure the accuracy and stability of the measurement conditions. Table 1 lists the
possible sources of uncertainty, which could affect the performance of the HTDMAs. In our setup,
nanoparticle sizing, aerosol/sheath flow rates, the high voltage (HV) applied to nano-DMAs, RH sensors,
and temperature sensors are calibrated and verified independently.
Note that in the following, for calibration and/or checking of different parameters, the criteria and/or
standard that the nano-HTDMA system has to meet are listed mainly according to the suggestions from
Duplissy et al., (2009) and Wiedensohler et al. (2012), which are not specifically provided for accurately
measuring sizes or hygroscopic growth of sub-10 nm nanoparticles. Compared with these criteria, to
measure the hygroscopic growth of sub-10 nm nanoparticles, we have achieved a better condition for our
nano-HTDMA system after comprehensive calibrations described as follows (more details about the
performance of our system see section 3).

### 2.2.1 Sizing accuracy

For particle diameters higher than 100 nm, the verification of sizing accuracy of DMAs can be
accomplished by using certified particles of known sizes such as polystyrene latex (PSL) spheres (Hennig
et al., 2005; Mulholland et al., 2006; Duplissy et al., 2009; Wiedensohler et al., 2012, 2018). The particle
sizing of nano-DMA2 is checked with PSL by switching off the sheath flow and the HV supply of nano-
DMA1, which actually in this case does not function as a DMA, but rather a stainless-steel tube. Sizing
agreement between measured diameters and nominal diameters of PSL particles above 100 nm should be
within ±3% (Wiedensohler et al., 2012). After confirming the accurate sizing of nano-DMA2, the sizing
accuracy of nano-DMA1 can be in turn checked by the nano-DMA2 with a full scan of a certain size of
PSL selected by the nano-DMA1. Note that, it is important to check not only the sizing accuracy of both
DMAs, but also the sizing agreement between the nano-DMA1 and nano-DMA2. To achieve good
hygroscopicity measurements of nanoparticles, the sizing offset of the two DMAs should be within ±2-3%
(Massling et al., 2011; Zhang et al., 2016).
For nanoparticles with diameters smaller than 100 nm, the sizing accuracy is, however, difficult to check
by using PSL nanoparticles. This is mainly because the size of residual material in the solution also peaks
around 20 − 30 nm (Fig. S2a), resulting in an asymmetric number size distribution of generated PSL
nanoparticles (Fig. S2b) (Wiedensohler et al., 2012). PSL nanoparticles with diameters below 20 nm are
not commercially available (https://www.thermofisher.com/order/catalog/product/3020A), making the
verification in this size range even impossible. Sizing accuracy of nanoparticles is critically determined by
sheath flow rates and HV applied to the nano-DMAs. However, unlike for the 100 nm nanoparticles, a ±2-
3% sizing offset between the two DMAs would be very difficult to maintain for nanoparticles with
diameters smaller than 20 nm. Thus, accurate calibrations of sheath flow rates and HV are crucial for

constraining the uncertainty associated with sizing of nanoparticles below 100 nm. The calibrations for aerosol/sheath flow, DMA voltage, and sensors will be described in detail in the following Section 2.2.2-2.2.5.

**2.2.2 Aerosol and sheath flow**

Sizing accuracy of a DMA directly depends on the accuracies of aerosol and sheath flow rates. The aerosol flow rate at the inlet of the nano-DMA1 is checked by using a bubble flow meter (Gilian Model Gilibrator-2). Wiedensohler et al. (2012) recommended that the measured aerosol flow rate should not deviate more than 5% from the set flow rate during the measurements, otherwise one should check the flow rate of CPC or if there is a leakage in the system. Details about leakage checking can be found in Birmili et al. (2016).

To calibrate the sheath flow, a verified MFM (TSI Series 4000) is placed in the recirculating sheath flow close-loop upstream of the MFM. By applying a series of sheath flow rates, a calibration curve (flow rate vs. MFM analog output) can be obtained according to the reading of the reference MFM. A maximum deviation of 2% from the sheath flow rate value of the reference MFM is recommended by Wiedensohler et al. (2012), which can keep sizing accuracy of 200 nm PSL particles within ±2%.

**2.2.3 DMA voltage**

The sizing of nano-DMAs is very sensitive to the accuracy and precision of the voltages applied, especially when measuring nanoparticles in the sub-10 nm diameter range. A verified reference voltage meter with voltage up to 1000 V (Prema Model 5000 DMM, accuracy 0.005%) is used to calibrate the HV supply of the nano-DMAs (0-350 V). By setting a series of analog voltage values, the HV applied to nano-DMA can be calibrated according to the values shown in the reference voltage meter. For our nano-DMAs, sub-10 nm in particle sizes correspond to voltage below 50 V. Thence, voltage calibration should be performed with a higher resolution (smaller voltage interval) from 0 to 50 V (shown in the inset of Fig. 2).

**2.2.4 RH sensor**

One typical method to calibrate RH sensors in a HTDMA system is to measure the hygroscopic growth factors of ammonium sulfate (Hennig et al., 2005), although the effects of shape factors, restructuring, and impurities in the solutions may hamper a reliable RH calibration with this method (Duplissy et al., 2009). Moreover, this indirect RH sensor calibration through measurement of the hygroscopic growth factors of ammonium sulfate (usually with nanoparticle diameters around or above 100 nm) only calibrates the RH values higher than the ERH of the pure salt. Calibration of RHs below ERH of ammonium sulfate is important for the phase transition measurements. Most importantly, we are investigating the hygroscopic growth factors of ammonium sulfate nanoparticles. Hence, using ammonium sulfate nanoparticles to calibrate RH sensors in our system becomes invalid.

Therefore, we alternatively calibrate the RH sensors by using a DPM (EDGE TECH Model MIRROR-99), which is recommended in several previous studies (Hennig et al., 2005; Duplissy et al., 2009; Biskos et al., 2006a, b, 2007). In the calibration, the DPM and RH sensors should be kept in the well-insulated chamber with constant laboratory conditions (e.g., flow rates, temperature, and pressure). By running the DPM and all the other RH sensors in parallel at various RHs (5% to 90%), a calibration curve of the RHs measured by the DPM against analog voltages of RH sensor can be obtained.

**2.2.5 Temperature sensor**

Since all our temperature sensors and the highly accurate DPM (EDGE TECH Model MIRROR-99) are installed in the aforementioned well-insulated chamber and the chamber temperature is maintained with air conditioner at about 292.15±0.1 K, we calibrate the temperature sensors and correct their systematic shift by comparing the record of temperature sensors and the DPM by keeping them in parallel inside the chamber over a 12-hour time period.

**2.3 Particle generation**

The experiments shown in this study were conducted using laboratory generated ammonium sulfate and
sodium sulfate nanoparticles. Nanoparticles with diameters of 6, 8, and 10 nm were generated by an
electrospray (AG: TSI Model 3480) with 1, 5, and 20 mM aqueous solution of ammonium sulfate and
sodium sulfate (Aldrich, 99.99%), respectively. The generated particles were then diluted and dried to RH
below 2% by mixing with dry and filtered $N_2$ (1 l/min) and $CO_2$ (0.1 l/min). The dried polydisperse aerosol
nanoparticles were subsequently neutralized by a $Po^{210}$ neutralizer. To avoid blocking the 25-μm capillary
of the electrospray with high solution concentration, we used an atomizer (AG: TSI Model 3076) to
generate nanoparticles with diameters of 60-100 nm and 20 nm with 0.05 and 0.001 wt% solution of
ammonium sulfate and sodium sulfate (Aldrich, 99.99%), respectively. Also, 100-nm PSL nanoparticles
were atomized from a PSL solution of mixing 3 drops of 100-nm PSL with 300 mL distilled and de-ionized
milli-Q water. The generated nanoparticles were subsequently dried to RH below 10% with a custom-built
Nafion dryer (ND: TROPOS Model ND.070) and then neutralized by a $Kr^{85}$ neutralizer.
The solutions used in our measurements were prepared with distilled and de-ionized milli-Q water
(resistivity of 18.2 MΩ cm at 298.15 K). Note that, for 100-60 nm and 20 nm, the solution concentration
was adjusted so that the sizes selected by the nano-DMA1 were always larger than the peak diameter of the
number size distribution of the generated nanoparticles to minimize the influence of the multiple charged
nanoparticles in hygroscopicity measurements. The influence of multiple charges on sub-10 nm particles
is expected to be very small, we, however, still used different concentrations so that the sizes selected by
the nano-DMA1 were always around the peak of the number size distribution of the generated nanoparticles
by the electrospray (Fig. S3). This is to ensure that we could have as many particles as possible to
compensate the strong loss of very small particles in the whole humidification systems.

**3 Results and discussion**
**3.1 Performance of the nano-HTDMA**

### 3.1.1 Sizing accuracy

In this section, we show the performance of our nano-HTDMA after a full calibration, including accuracy and stability of the aerosol/sheath flow rates, the voltage applied to the nano-DMAs, and nanoparticle-sizing accuracy. In our study, the sheath/aerosol flow rates and nano-DMA voltage supply have been calibrated every day and every two weeks, respectively. The deviations of the measured aerosol/sheath flow rates from the set-point values are less than ±1%, which is lower than the maximum variation of 2% recommended by Wiedensohler et al. (2012).

The voltage applied to the nano-DMAs (up to 350 V) is kept within ±0.1% around the set value shown in the voltage meter. As shown in Fig. 3a, when test with 100-nm PSL nanoparticles, the average peak diameter of scans from the nano-DMA2 is 100.4 nm, which matches well with the mean diameter of PSL nanoparticles (100±3 nm, Thermo Fisher Scientific Inc.). Afterwards, when using nano-DMA1 to select 100 nm PSL, the scanned size distribution by nano-DMA2 has a peak diameter at 100.3 nm (Fig. 3b), indicating a good sizing accuracy of the nano-DMA1 too. As discussed in Sec. 2.2.1, it is difficult to verify the sizing accuracy of sub-100 nm aerosol nanoparticles using PSL nanoparticles. Duplissy et al. (2009) and Wiedensohler et al. (2012) suggested estimating the sizing accuracy of sub-100 nm nanoparticles through the DMA transfer function. The theoretical DMA transfer function (see SI. S2. Eq. (S2-S4)) was proposed by Knutson and Whitby (1975) and they noted that sizing is crucially dependent on flow rates and HV applied to the DMA. In our study, the flow accuracy calibrated by the mass flow meter (TSI series 4000) is within ±2%. The variation of voltage applied to the nano-DMAs (0-12500 V, 0-350 V) around the set value was measured with voltage power supply (HCE 0-12500, HCE 0-350, Fug Electronic) and summarized in Table S5. According to the error propagation formula (see SI. S2. Eq. (S5)) (Taylor and Taylor, 1997), the calculated uncertainty in the sizing of 6-100 nm nanoparticles increases as size decreases (Table S5). The estimated sizing accuracy is slightly smaller than the sizing offset of two nano-DMAs, but in principle they are still consistent with each other. This suggests that uncertainties of slip correction, DMA

dimensions (inner and outer radius, length), temperature, pressure, and viscosity of air may also affect the sizing accuracy (see SI. S2. Eq. (S4), Kinney et al., 1991). Besides, Wiedensohler et al. (2012) also suggested that particle losses, the size- and material-dependent CPC counting efficiency can affect the size accuracy of DMAs.

After calibration, on average a <1.4% sizing offset between the two nano-DMAs can be achieved for ammonium sulfate nanoparticles with dry diameters of 100 nm, 60 nm and 20 nm (Fig. 3c, Fig.5, Table S3, Fig. S4, and Fig. S5), which is much better than the 2-3% criteria recommended by Massling et al. (2011) and Zhang et al. (2016). For sub-10 nm ammonium sulfate nanoparticles, our system has an average sizing offset of <0.9% for 10 and 8 nm particles and ~1.4% for 6 nm particles, respectively (Fig. 3d, Fig. 5, Table S3, and Fig. S6). As discussed above, uncertainties in the sheath flow rates and nano-DMA voltages will increase as size decreases, which results in a larger sizing offset of 6-nm nanoparticles compared with other sizes. Note that, we also tested the calibration of the DMA voltage with a voltage meter with lower accuracy of ±1%, and the DMA voltages can only be kept within ±1% around the set value. In this way, we found a much larger sizing offset for the sub-10 nm particles, i.e., 5.4% and 6.0% for 8 and 6 nm ammonium sulfate nanoparticles, respectively. These results show that maintaining an accurate sheath/aerosol flow (with ±1% around the set value) together with a careful voltage calibration (with ±0.1% around the set value, especially in low voltage range, i.e., <50 V for our system) is the key for accurate sizing of sub-10 nm nanoparticles.

**3.1.2 Preventing pre-deliquescence in the deliquescence measurement mode**

Pre-deliquescence of dry nanoparticles in the deliquescence measurement mode is an important issue that needs to be resolved in order to obtain accurate DRH (Biskos et al., 2006b; Duplissy et al., 2009; Bezantakos et al., 2016; Hämeri et al., 2000). Since temperature and RH are closely linked and accurate monitoring of these two quantities in the system is critical for nano-HTDMA measurements, we calibrated all RH and T sensors regularly (every two weeks in this study). To prevent pre-deliquescence and optimize the system, we have conducted three tests using ammonium sulfate nanoparticles with a dry diameter of

100 nm. In the first test, we regulated the RH of excess flow ($RH_e$) and made it equal to that of the aerosol
flow at the inlet of nano-DMA2 ($RH_a$), i.e., $RH_e=RH_a$, as done by previous HTDMA measurements, e.g.,
Villani et al. (2008). As shown in Fig. 4a, the measured growth factors of 100-nm ammonium sulfate are
in good agreement with predictions of the Extended Aerosol Inorganic Model (E-AIM; Clegg et al., 1998)
at RH above 80%. However, the ammonium sulfate nanoparticles deliquesce at 75% RH, which is
significantly lower than the expected DRH (80%, Tang and Munkelwitz (1994)). Since our RH sensors
were all well calibrated and the uncertainty of RH measurement is ±1%, it is reasonable to hypothesize that
the RH upstream of nano-DMA2 has already reached the deliquescence RH of ammonium sulfate
nanoparticles. When these aerosol nanoparticles move downstream of the nano-DMA2, the RH decreases
back to 75%, which dehydrates the deliquesced ammonium sulfate nanoparticles. To avoid the pre-
deliquescence, Hämeri et al. (2001) has suggested to set $RH_a$ to be 3-5% lower than $RH_e$. In the second test,
we have configured and regulated the system following this suggestion, i.e., $RH_e \geq RH_a+3\%$. In this case,
the ammonium sulfate nanoparticles still deliquesce at 79% RH (Fig. 4b), even if $RH_a$ is 6% lower than
$RH_e$.
Previous studies (Biskos et al., 2006b; Bezantakos et al., 2016) have shown that RH non-uniformities within
the nano-DMA2 can result in inaccurate measurements of phase transition and hygroscopic growth of
aerosol nanoparticles. One reason for RH non-uniformities within nano-DMA2 is that the sheath flow RH
is different from the aerosol flow RH at the inlet of the DMA (Hämeri et al., 2000, 2001). Another important
reason is the existence of temperature gradient within nano-DMA2 (Bezantakos et al., 2016). Hence, in the
third test, we moved the RH sensor from the excess flow downstream of nano-DMA2 to the sheath flow
upstream of nano-DMA2 and then regulated RH of sheath flow ($RH_s$) the same as $RH_a$ (shown in Fig. 1),
i.e., $RH_s=RH_a$, as done by Kreidenweis et al. (2005), Biskos et al. (2006a, b), and Massling et al. (2011).
Note that to minimize the temperature gradient within the nano-DMA2 in our system so that nanoparticles
can undergo almost the same RH conditions, the nano-DMA2 with its sheath flow humidification system
has been placed in a well-insulated air-conditioned chamber. The air temperature inside the chamber can

be maintained at an almost constant level (292.15±0.1 K). In addition, a heat exchanger was installed downstream of the recirculation blower to minimize the temperature perturbation in the sheath flow by the heat generated in the RB. Unlike previously reported by Bezantakos et al. (2016) that the RH at the outlet was higher than that the inlet of the sheath air, we monitored that the sheath flow temperature at the inlet of nano-DMA2 is slightly lower (less than ~0.2 K) than that at the outlet (i.e., the $RH_s$ at the inlet of nano-DMA2 is slightly higher (~ 1%) than the RH of the excess air at the outlet), while temperature of sheath flow is equal to that of aerosol flow at the inlet of nano-DMA2 during the measurements. A small temperature difference within nano-DMA2 is more likely due to the heat transfer between the inner electrode and air which flows around it by convection/conduction (Bezantakos et al., 2016). The plausible reason could be that when charged nanoparticles (similar to the electric current) hit the inner electrode, the inner electrode has some resistive heating from the electric current that flows. Such temperature difference/gradient within DMA was observed by previous studies (Biskos et al., 2007; Villani et al., 2008; Dupplissy et al., 2009; Bezantakos et al., 2016; Giamarelou et al., 2018). For example, a ±0.5 °C temperature difference within DMA was observed by Giamarelou et al. (2018) during the measurements. Except for the possibly slightly higher temperature of the inner electrode than the surrounding air, temperature gradient in DMA2 may also be caused by environmental disturbance or temperature difference between other parts of DMA and between sheath flow and aerosol flow. In this study, we calculate the change in heat ($Q$) of a nano-DMA2 system at a constant pressure, which estimates to be ~0.08 W ($Q = mdTC_{p,k}$) by considering the density and heating capacity of air, and aerosol and sheath air flow rate ($\rho$=1.2041kg/m$^3$; $C_p$= 1.859kJ/kg°C) (Atkins et al., 2006). Although this temperature perturbation (less than ~0.2 K between the sheath flow at the inlet and the excess flow at the outlet of the nano-DMA2) is larger than the ideal condition of less than 0.1 K that Duplissy et al. (2009) and Wiedensohler et al. (2012) suggested, our experimental results show that a prompt phase transition can be still achieved. In this case, the measured DRH of ammonium sulfate nanoparticles is almost at 80% (Fig. 4c and 4d).

### 3.1.3 Prompt phase transition of ammonium sulfate

Figure 5 and 6 show the normalized particle number size distributions measured by the nano-DMA2 in the
respective deliquescence and efflorescence measurement modes for ammonium sulfate nanoparticles with
dry mobility diameters of 20 nm, 10 nm, and 6 nm (see Fig. S4 for 100 nm, see Fig. S5 for 60 nm, see Fig.
S6 for 8 nm). In the deliquescence measurement mode (Fig. 5, Fig. S4a, and Fig. S5a), we observed a
similar double-mode phenomenon as reported by Mikhailov et al. (2004) and Biskos et al. (2006b, 2007).
For example, at 20 nm, there are two distinct intersecting modes of particle size distributions determined
by the nano-DMA2 in the RH range from 79% to 83% RH (around the DRH of ammonium sulfate). Biskos
et al. (2006b, 2007) attributed these two modes to the co-existence of solid and liquid phase nanoparticles
at RH close to the DRH of ammonium sulfate, due to the slight inhomogeneity of RH in the second nano-
DMA, i.e., some nanoparticles have already undergone deliquescence (liquid state) and some are not
(solid). This is evident through a double-mode log-normal fitting (red and blue modes in Fig. 5). Until RH
~82%, the peak diameter of the red mode at 82% RH is similar to that at 11% RH, indicating that these
nanoparticles are still in a solid state. At 82% RH, a population of ammonium sulfate nanoparticles starts
to deliquesce and exists in a distinct mode with significantly larger peak diameter (blue mode), although a
majority of the nanoparticles remain solid (red mode). As RH further increases, the peak diameter of the
normalized number size distribution of the blue mode increases, indicating the continuous growth of the
nanoparticles after deliquescence. However, in our case, the double-mode phenomenon was not observed
for 8 and 6 nm ammonium sulfate nanoparticles (Fig. 5 and Fig. S6a). To have a better estimation of DRH
when the double modes occurred, the peak diameter of the mode with the larger number of nanoparticles
was chosen for growth factor calculation (Biskos et al., 2006b, 2007). For example, for 20 nm ammonium
sulfate nanoparticles, the peak diameters of the normalized number size distribution of the red and blue
modes are used to calculate growth factor at RH between 79% to 83%, respectively.
For the efflorescence measurement mode, we adopted the approach of Biskos et al. (2006b) and used the
geometric standard deviation of number size distribution (sigma: σ) to quantify the diversity of the sizes of
nanoparticles. As shown in Fig. 6, Fig. S4b, Fig. S5b, and Fig. S6b, broadening of the normalized number
size distributions measured with nano-DMA2 was only observed for 20-nm ammonium sulfate
nanoparticles in the RH range from 33% to 30%. There, at RH higher than 33% or lower than 30%, σ stays
stably at 1.072. However, clear increases of σ (1.078-1.087) were observed for RH between 33% and 30%.
The normalized number size distributions in the RH range from 33% to 30% can be further resolved by
double-mode fit with a fixed σ of 1.072 (the red and the blue mode in Fig. 6 for 20 nm). The ammonium
sulfate nanoparticles in the red mode at RH between 33% to 30% are in the solid state because the peak
diameter of red mode is similar to that at 11% RH. However, within this RH range, the peak diameter of
the blue mode is significantly larger, indicating that these nanoparticles are still in the liquid state. Further
decreasing RH (lower than 30%), only one mode has been observed and the peak diameter of the normalized
number size distribution almost unchanged as RH decreases (red mode in Fig. 6 for 20nm), which means
that the nanoparticles have been all in the solid state. Similar to the deliquescence measurement shown
above and in Fig. 5, the co-existence of solid and aqueous phase nanoparticles at RH 30-33% is also very
likely to stem from the slight heterogeneous RH in nano-DMA2 (Biskos et al., 2006b). To have a better
estimation of ERH when the broadening phenomenon exists, the peak diameter of the mode with the larger
number of nanoparticles was used for growth factor calculation. After such data processing in both
deliquescence and efflorescence modes, we obtained prompt deliquescence and efflorescence of 6 to 100
nm ammonium sulfate nanoparticles (more details in Section 3.1.4).
**3.1.4 Size-dependent hygroscopicity of ammonium sulfate nanoparticles**
Figure 7 shows the humidogram of ammonium sulfate nanoparticles measured by our nano-HTDMA
system in the size (dry diameter) range of 6-100 nm. The detailed comparison between our results and
Biskos et al. (2006b) during both deliquescence and efflorescence measurements are presented in Fig. 8a
and b (also Fig. S7). In general, our results are in good agreement with the measurement results of Biskos
et al (2006) and the theoretical prediction by Cheng et al. (2015). First, there is a strong size dependence in
the hygroscopic growth factor of ammonium sulfate nanoparticles, and smaller ammonium sulfate
nanoparticles exhibit lower growth factor at a certain RH. For example, the difference in the growth factor
between 6 and 100 nm nanoparticles is up to 0.28 at 80% RH (Fig. S8a). Second, there is, however, no
significant size dependence in both DRH and ERH (Fig. S8b). For nanoparticles of different sizes (6-100
nm), the DRH and ERH of ammonium sulfate vary slightly from ~80-83% and ~30-34%, respectively. This
variation of the DRH and ERH along the size is much smaller for ammonium sulfate nanoparticles than for
sodium chloride (Biskos et al. 2006a, 2007).
Although our results in general agree well with Biskos et al. (2006b), the growth factors of 10, 8, and 6 nm
ammonium sulfate nanoparticles that we measured at high RH (i.e., > ~70%) are slightly lower (~0.02 in
growth factor) than that in Biskos et al. (2006b) in both deliquescence and efflorescence processes (Fig. 8b
and Fig. S7). We calculated the uncertainties of growth factor of 10-nm ammonium sulfate from 80% to
90% RH for our system and Biskos et al. (2006b) system by $\sqrt{\left(\left(g_f \frac{\sqrt{2}\varepsilon_{Dp}}{D_p}\right)^2 + \left(\varepsilon_{RH} \frac{dg_f}{dRH}\right)^2,\right)}$ (Mochida
and Kawamura, (2004)).  Here, $\varepsilon_{Dp}$, $\varepsilon_{RH}$, and $g_f$ are uncertainty of particle mobility diameter, uncertainty
of relative humidity, and growth factor with respect to RH, respectively. The average sizing offsets of our
system and Biskos et al. (2006b) for 10 nm ammonium sulfate are taken here as $\frac{\varepsilon_{Dp}}{D_p}$ (see Table 1). As shown
in the inset of Fig. 8b, the discrepancies between the two systems are still within measurement uncertainty.
In addition, compared to Biskos et al. (2006b), our results show a similar re-structuring in deliquescence
mode at RH between about 20% to 75% for 100, and 60 nm ammonium sulfate nanoparticles (Fig 8c).
However, different than in Biskos et al. (2006b), we do not find re-structuring for smaller ammonium
sulfate nanoparticles (20, 10, 8, and 6 nm) at RH below deliquescence point (Fig. 8c and Fig. 8d). There
seems to be continuous water adsorption and the adsorbed water layers (Romakkaniemi et al., 2001)
become significantly thicker when RH closer to the DRH (i.e, RH > 70%). For example, a slight increase
in the hygroscopic growth factor of 6-nm ammonium sulfate nanoparticles is observed in the RH range
from 65 to 79% RH before deliquescence. This is attributed to water adsorption onto the surfaces of these
nanoparticles. It seems that smaller nanoparticles have a stronger tendency of adsorbing water when
approaching the DRH than the larger ones. A similar phenomenon has also been observed by Hämeri et al.
(2000, 2001), Romakkaniemi et al. (2001), Biskos et al. (2006a, b, 2007), and Giamarelou et al. (2018).
The reason for such enhanced adsorption at smaller sizes is still to be investigated. Note that, the ammonium
sulfate hygroscopic data from Biskos et al. (2006b) shown here are all generated by an electrospray, but in
our experiments, only the ammonium sulfate nanoparticles with diameters smaller than 20 nm (i.e., 10, 8,
and 6 nm) were generated by an electrospray, while the larger nanoparticles (i.e., 20, 60, and 100 nm) were
generated by an atomizer. Different from generation conditions of for 6-10 nm ammonium sulfate
nanoparticles in Biskos et al. (2006b), in our study, in order to minimize the multiple charged nanoparticles,
three different concentrations are used so that the size selected by the nano-DMA1 (i.e., 6, 8, 10 nm) was
always slightly larger than peak of the number size distribution of the generated nanoparticles by the
electrospray. This also helps us to have as many as nanoparticles as possible to compensate the strong
nanoparticle losses in the nano-HTDMA system. In addition, we used both electrospray and atomizer to
generate 20-nm ammonium sulfate and compared their hygroscopic growth factors prior to deliquescence.
Figure S12a shows a ~ 0.1 higher growth factor of 20 nm ammonium sulfate generated by the electrospray
than that using the atomizer in the RH range from 55% to 82%, which is similar to the difference in the
hygroscopic growth factor of 20 nm NaCl aerosol nanoparticles using the different generation method as
observed in Fig S12b in Biskos et al. (2006a). Besides different generation conditions, the morphology of
dried ammonium sulfate particles may also differ slightly between our study and Biskos et al. (2006)
because of different dying rate, as drying flow rates and RH of the dried ammonium sulfate in the two
HTDMA systems are different too. This means the different generation methods and drying conditions may
influence the surface structure of the nanoparticles and thus their interaction with the adsorbed water layers
(Iskandar et al., 2003; Xin et al., 2019).
**3.2 Size-dependent hygroscopicity of sodium sulfate nanoparticles**
As a common constituent of atmospheric aerosol particles (Tang and Munkelwitz, 1993, 1994; Tang 1996;
Tang et al., 2007), the hygroscopicity of sodium sulfate with diameters above 20 nm particles has been
investigated by a few groups (Tang et al., 2007; Xu and Schweiger, 1999; Hu et al., 2010). However, its
hygroscopic behavior in the sub-10 nm size range has not been investigated yet. In this study, we applied
our nano-HTDMA system to measure the hygroscopic growth factors, DRH, and ERH of sodium sulfate
nanoparticles with dry size from 20 nm down to 6 nm.
Figure 9 shows the measured size-resolved hygroscopic growth factors of sodium sulfate nanoparticles.
Different from the observations by Tang et al. (2007) using an electrodynamic balance (EDB), we observed
prompt deliquescence and efflorescence for both 20-nm and 6-nm sodium sulfate nanoparticles. Two
intersecting modes in the measured number size distribution of humidified sodium sulfate nanoparticles are
observed at RH close to the DRH (Fig. S9 and S10 in the Supplementary Information) and ERH, suggesting
an external mixture of aqueous and solid nanoparticles. As shown in Sect. 3.1.3, a similar phenomenon is
also observed for ammonium sulfate, which could be attributed to the slight RH heterogeneities in nano-
DMA2, which makes only part of the nanoparticles deliquesce at RH close to the DRH, while the others
remain in the solid state.
Together with the hygroscopic growth of 14-16 µm and 200-20 nm sodium sulfate measured previously by
Tang et al. (2007) and Hu et al. (2010), we show a strong size dependence in hygroscopic growth factors
of sodium sulfate nanoparticles (Fig. S11d).  For example, at RH=84%, the hygroscopic growth factor of 6
nm sodium sulfate is only ~ 1.3 (in efflorescence mode), while the respective growth factors are about 1.5
and 1.8 for 20 nm and 14-16 µm particles. As shown in Fig. 9, E-AIM already agrees well with the
hygroscopic growth of micrometer particles (14-16 µm) without shape correction (DeCarlo et al., 2004),
i.e., shape factor ($\chi$) of 1.0. However, to explain observation, a shape factor of ~1.16 and 1.26 would be
needed for 20 nm and 6 nm sodium sulfate nanoparticles, respectively.
There is no significant change in DRH between 14-16 µm (~84%) and 20 nm (~84%) sodium sulfate
particles (Fig. 9). This is consistent with Hu et al. (2010) where no change in DRH from 200 nm down to
20 nm (~82%, see Table 1 from Hu et al. (2010)) was observed. However, a significant increase of DRH
occurred when further decreasing particle diameters to 6 nm (DRH = ~90%). The size dependence of ERH
is stronger than that of DRH, as there is already a clear increase of ERH from micrometer 14-16 µm (~57%)
to 20 nm (~62%) sodium sulfate particles. When further reducing the particle diameters to 6 nm, an almost
6% increase of DRH can be found, compared to the micrometer 14-16 µm particles (i.e., ERH increases
from 57 to 82%, respectively). Different from ammonium sulfate, of which DRH and ERH show no
significant size dependence, there is a strong size-dependence of DRH and ERH of sodium sulfate
according to our observations down to 6 nm. The different size dependence of DRH and ERH between
sodium chloride and ammonium sulfate have been theoretically studied and explained by Cheng et al.
(2015). The main reason is the different concentration dependence of solute activities and the different
solute-liquid surface tension, e.g., the same change in solute molality leads to a larger change in the solute
activity of sodium chloride than that of ammonium sulfate. The phase transition concentration
(deliquescence and crystallization concentration) of ammonium sulfate is thus more sensitive to the size
change compared to that of sodium chloride, leading to the almost unchanged DRH and ERH of ammonium
sulfate nanoparticles (Cheng et al., 2015). For the size dependence of phase transition of sodium sulfate, a
strong size effect on DRH and ERH is similar to that of sodium chloride but different from that of
ammonium sulfate in the size range from 6 to 20 nm, suggesting that non-ideality of solution property is
close to that of sodium chloride but weaker than that of ammonium sulfate. As different hydrates of sodium
sulfate may exist during the deliquescence and efflorescence processes (Xu and Schweiger, 1999), to
explain the underlying mechanism of the size dependent hygroscopicity of sodium sulfate particles can be
challenging.

## 4 Summary and Conclusion

In this study, we presented our newly designed and self-assembled nano-HTDMA for measuring hygroscopicity of nanoparticles in the sub-10 nm diameter size range. We also introduced the comprehensive methods for system calibration and reported the performance of the system, focusing on the sizing accuracy and preventing pre-deliquescence in the deliquescence measurement mode. By comparing with previous studies on ammonium sulfate nanoparticles (Biskos et al., 2006b), we show that our system is capable of providing high quality data of the hygroscopic behavior of sub-10 nm nanoparticles. We then extended our measurements for sodium sulfate nanoparticles, of which size-dependent deliquescence and efflorescence have been clearly observed for nanoparticles down to 6 nm in size, with similar behavior as sodium chloride.

As we know, atmospheric aerosol particles consist of not only inorganic components, but also a vast number of organic components existing in the atmosphere. However, their physico-chemical properties are still not fully understood, especially when comes to the nano-scale and supersaturated concentration range. The nano-HTDMA system can be directly applicable to explore the size dependence of aerosol nanoparticles. Combing the multi-size measurements of hygroscopicity and the Differential Köhler Analyses (DKA, Cheng et al., 2015) in nano size range, we will be able to characterize and parameterize the water activity and surface tension of different inorganic and organic systems. This will further help us to understand the formation and transformation of aerosol nanoparticles in the atmosphere and their interaction with water vapor.

**Data availability**

Readers who are interested in the data should contact Yafang Cheng (yafang.cheng@mpic.de).

**Acknowledgement**

This study was supported by the Max Planck Society (MPG) and Leibniz Society. T.L. acknowledges the support from China Scholarship Council (CSC). Y.C. would like to acknowledge the Minerva Program of MPG.

**Author contributions**: Y.C. and H.S. designed and led the study. N.M., T.T. and A.W. assembled the basic HTDMA system. Y.C., H.S. and T.L. modified and advanced the basic system into the nano-HTDMA for the purpose of measuring hygroscopic properties of aerosol nanoparticles in sub-10 nm size range at MPIC. T.L. performed the experiments. J.H., N.M. and X.W. supported the experiments. All co-authors discussed the results and commented on the manuscript. T.L. wrote the manuscript with input from all co-authors.

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

# Tables

**Table 1.** Accuracy, precision and sources of uncertainty associated with HTDMA measurements.

|  | Biskos et al. (2006b) | Hämeri et al. (2000) | Nano-HTDMA (This study) |
|---|---|---|---|
| *DMA System* | | | |
| Type of DMA1 & DMA2 | TSI nano-DMAs | Hauke-type DMAs | Vienna-type short DMAs |
| Accuracy of aerosol flow in DMA2 | ±1% (0.3-1.5 l/min) | - | ±1% (1.5 l/min) |
| Accuracy of sheath flow in DMA2 | ±1% (5-15 l/min) | - | ±1% (10 l/min) |
| Accuracy of DMA voltage | ±0.1% (0-500V) | - | ±0.1% (0-350V) |
| Sizing accuracy of DMA2 using PSL | 3% | - | 0.4% (100-nm PSL) |
| Sizing agreement between DMAs using ammonium sulfate | 3.1% (10 nm) [a] | ±1% [b] | 0.6% (100 nm) [c] <br> 0.5% (60 nm) [c] <br> 1.4% (20 nm) [c] <br> 0.9% (10 nm) [c] <br> -0.2% (8 nm) [c] <br> 1.4% (6 nm) [c] |
| Precision of particle-sizing | <2% | - | <2% (6-200 nm) [d] |

| *Humidification System* | | | |
|---|---|---|---|
| Type of RH sensor | RH sensors<br><br>(Omega Model HX93AV) | Dew point mirror (GE)<br>RH sensors<br><br>(Vaisala Humitter model 50Y) | Dew point mirror (Edge)<br>RH sensors<br><br>(Vaisala model HMT 330) |
| Accuracy of RH sensors<br><br>(0-90% RH) | $\pm 2.5\%$ RH | $\pm 3\%$ RH [e] | $\pm 1\%$ (RH sensor) |
| Position of the probe in the system | Inlet of DMA2<br><br>($RH_a$ sensor [f], $RH_s$ sensor [g]) | Inlet of DMA2 ($RH_a$ sensor) & excess air<br><br>($RH_s$ sensor, dew point mirror) | Inlet of DMA2 ($RH_a$ sensor,<br><br>$RH_s$ sensor) & excess air<br><br>(dew point mirror) |
| RH setting | $RH_a=RH_s$ | $RH_s \geqslant RH_a+3\%$ | $RH_a=RH_s$ |
| *Temperature Control System* | | | |
| Temperature control type | Thermally isolated environment<br><br>(humidification+DMA2) [h] | Thermally isolated<br><br>environment (DMA2) | Box T regulated<br><br>(humidification+DMA2) |
| Difference in T between inlet and outlet of DMA2 | - | - | $<0.2°C$ |

- Not reported.
[a] According to the scans of the second DMA for the hygroscopic growth of 10 nm ammonium sulfate and the growth factors at different RHs provided by Biskos et al.
(2006b), we retrieved an average sizing offset of Biskos et al. (2006b) system to be ~3.1% at 10 nm (see SI, S1).
[b] Size range not given.
[c] See Table S2 in supporting information.
[d] Value calculated according to the relative standard derivation.
[e] From Vaisala Humitter model 50Y manual.
[f] $RH_a$: the RH of aerosol flow.
[g] $RH_s$: the RH of sheath flow.
[h] Bezantakos et al. (2016).
**Figures**

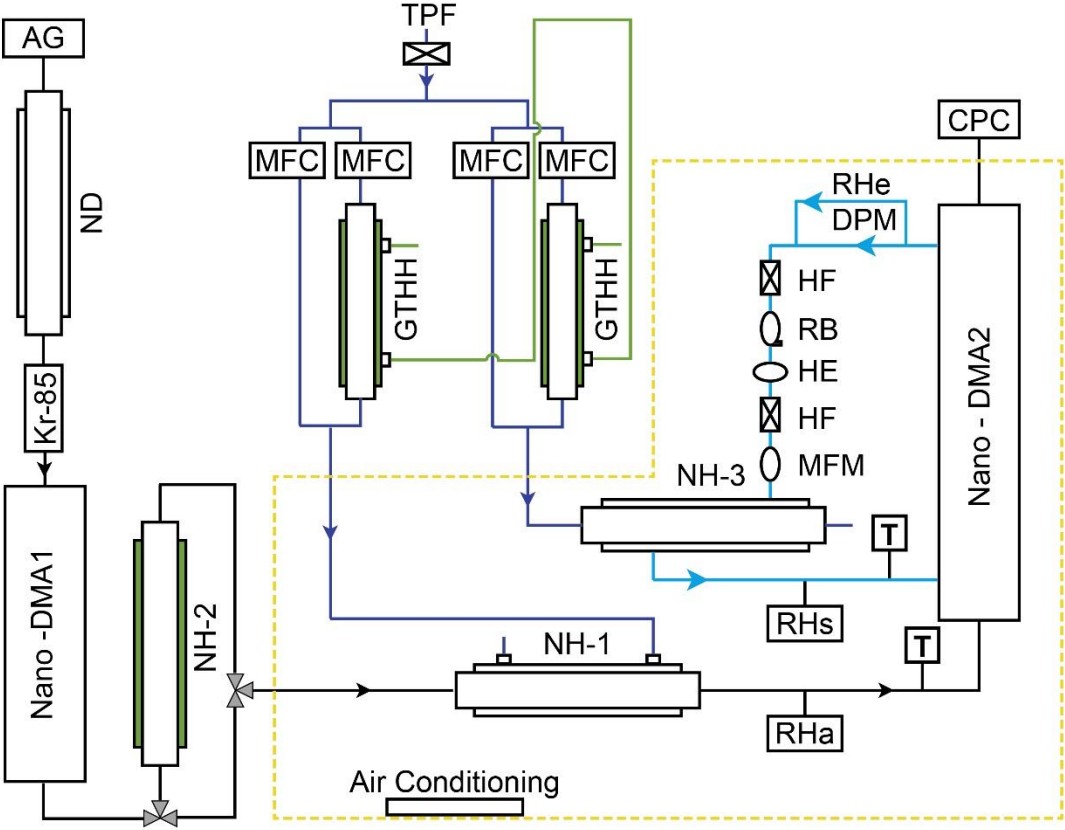


**Figure 1.** Experimental setup of the nano-HTDMA. Here, AG: aerosol generator (aerosol atomizer or electrospray);
ND: nafion dryer; Kr-85: Krypton source aerosol neutralizer; Nano-DMA: nano differential mobility analyzer; TPF:
total particle filter; HF: hydrophobic filter; MFC: mass flow controller; MFM: mass flow meter; RB: recirculation
blower; DPM: dew point mirror; GTHH: Gore-Tex humidifier and heater; NH: nafion humidifier; HE: heat exchanger;
CPC: condensation particle counter; Black line: aerosol line; Blue line: sheath line;  Royal blue line: humidified air;
Green line: MilliQ water (resistivity of 18.2 MΩ cm at 298.15 K). $RH_a$ and $RH_s$ (measured by RH sensors) represent
the RH of aerosol and sheath flow in the inlet of nano-DMA2, respectively. $RH_e$ (measured by dew point) represents
the RH of excess air. T represent the temperature of aerosol and sheath flow in the inlet of nano-DMA2, respectively.



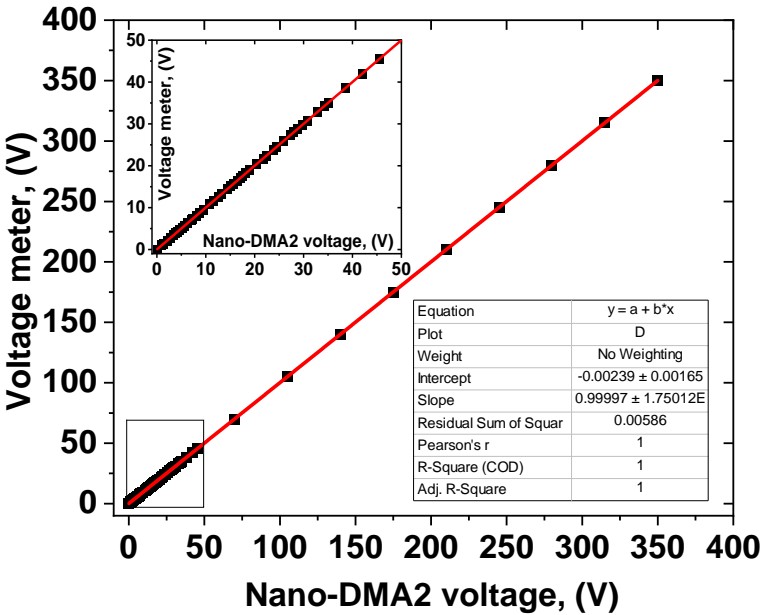

The following data table appears within the figure:

| Equation | y = a + b*x |
|---|---|
| Plot | D |
| Weight | No Weighting |
| Intercept | -0.00239 ± 0.00165 |
| Slope | 0.99997 ± 1.75012E |
| Residual Sum of Squar | 0.00586 |
| Pearson's r | 1 |
| R-Square (COD) | 1 |
| Adj. R-Square | 1 |


**Figure 2.** An example of voltage calibration of the nano-DMA2.








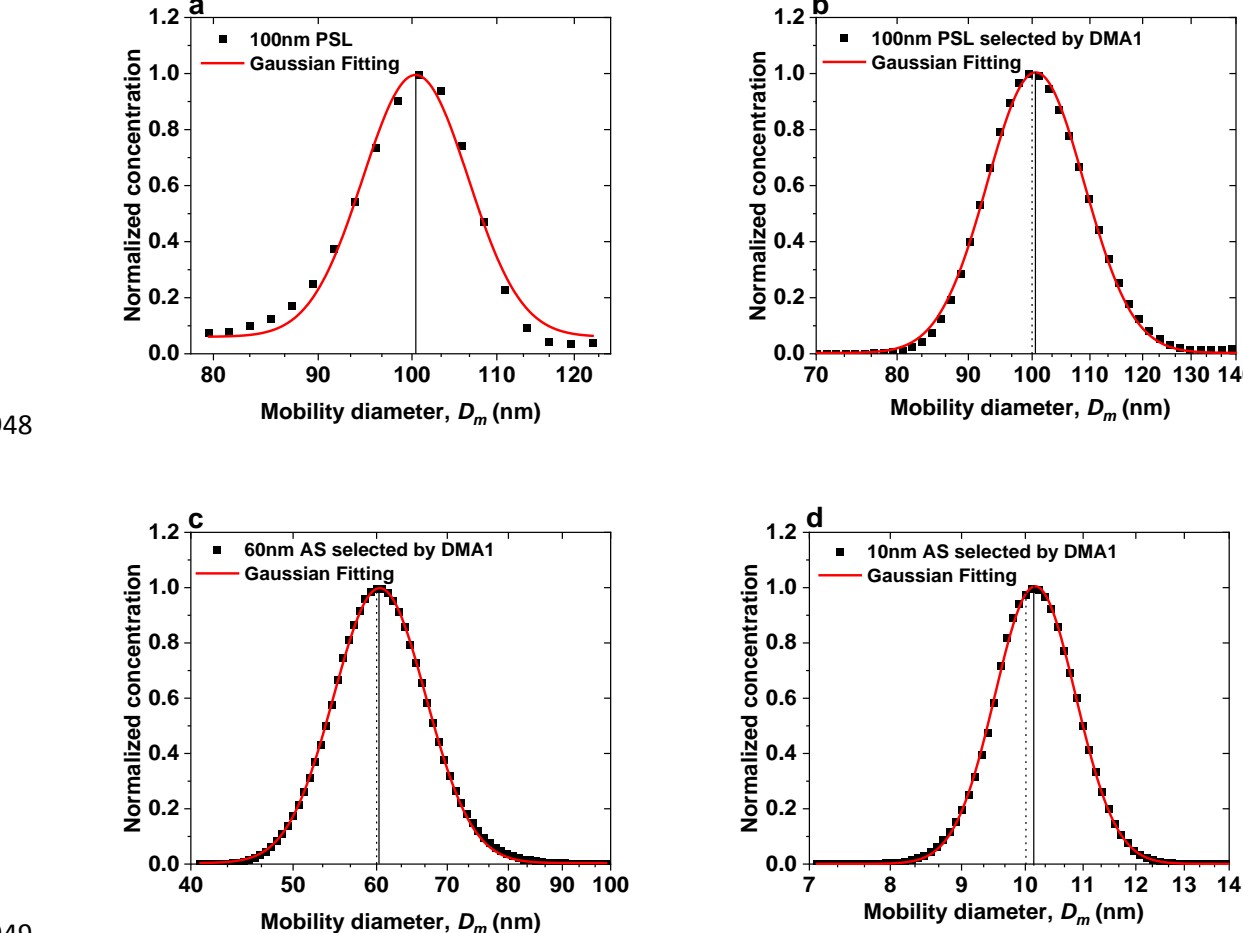



**Figure 3**. Sizing accuracy and sizing offset of nano-DMAs after calibration. **(a)** Normalized number size distribution scanned by the nano-DMA2 for 100-nm PSL nanoparticles (black solid square). Normalized number size distributions scanned by the nano-DMA2 for 100-nm PSL nanoparticles **(b)**, 60-nm **(c)**, and 10-nm **(d)** ammonium sulfate (AS) selected by the nano-DMA1 at RH below 5% at 298 K (black solid square). The dotted lines mark the diameters of the monodispersed nanoparticles selected by the nano-DMA1, i.e., 100 nm in **(b)**, 60 nm in **(c)** and 10 nm in **(d)**. The black solid lines mark the peak diameters from the Gaussian fits (red curve).

956

957

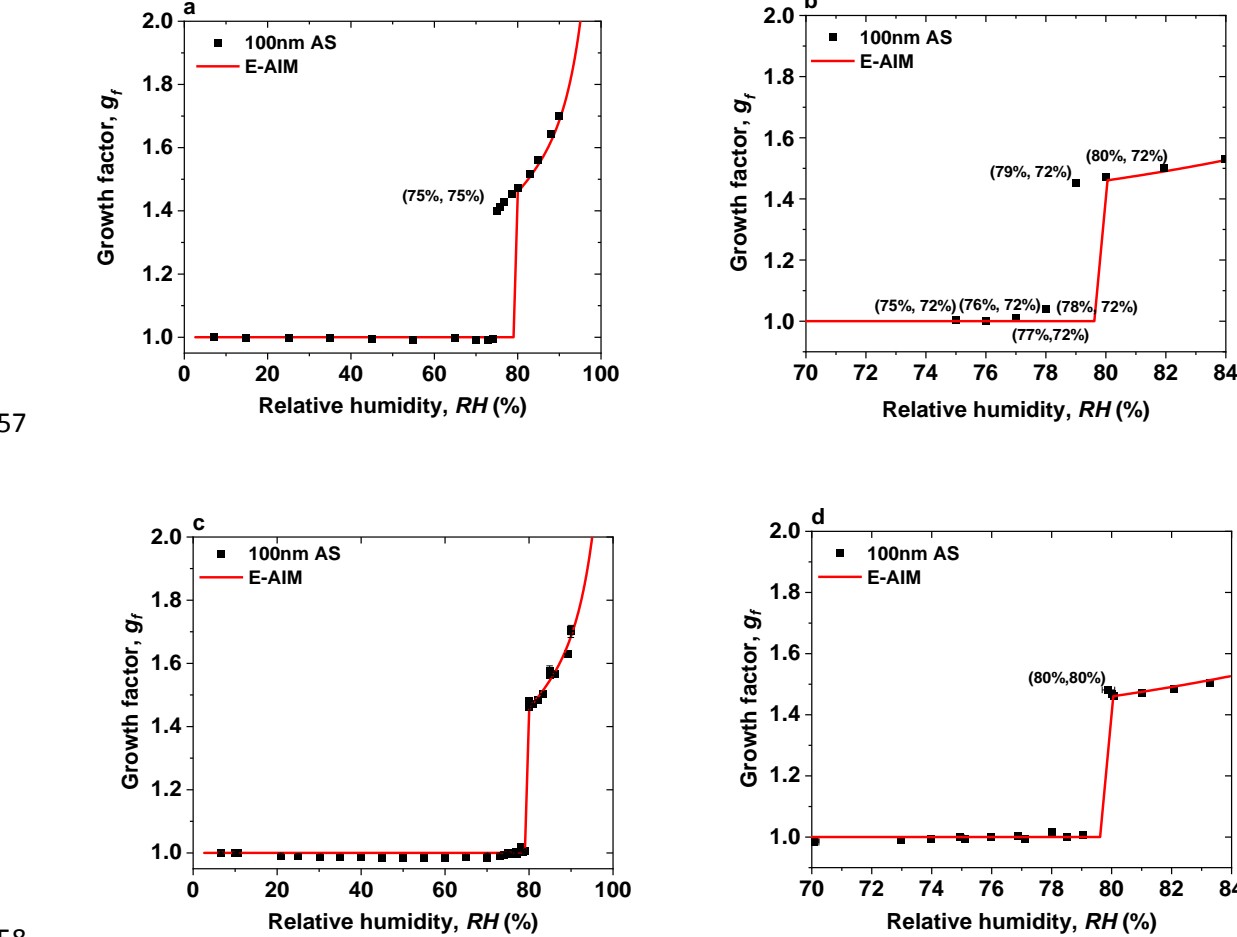

958

**Figure 4.** Mobility-diameter hygroscopic growth factors ($g_f$) of 100-nm ammonium sulfate (AS) nanoparticles at 298 K measured in deliquescence mode. In comparison, the E-AIM model predicted growth factors of ammonium sulfate nanoparticles at 100 nm. **(a) $RH_e=RH_a$,** (75%, 75%) represents the ($RH_e$, $RH_a$), **(b) $RH_e \geq RH_a+3\%$,** (75%, 72%) represents the ($RH_e$, $RH_a$), and **(c) $RH_s = RH_a$.** (**d**) The enlarged view of the RH range of 70% to 84% in Fig. 4c. (80%, 80%) represents the ($RH_s$, $RH_a$). $RH_s$ and $RH_e$ are the RH of sheath flow in the inlet of nano-DMA2 and in the excess air line, respectively; $RH_a$ is the RH of aerosol flow in the inlet of nano-DMA2.





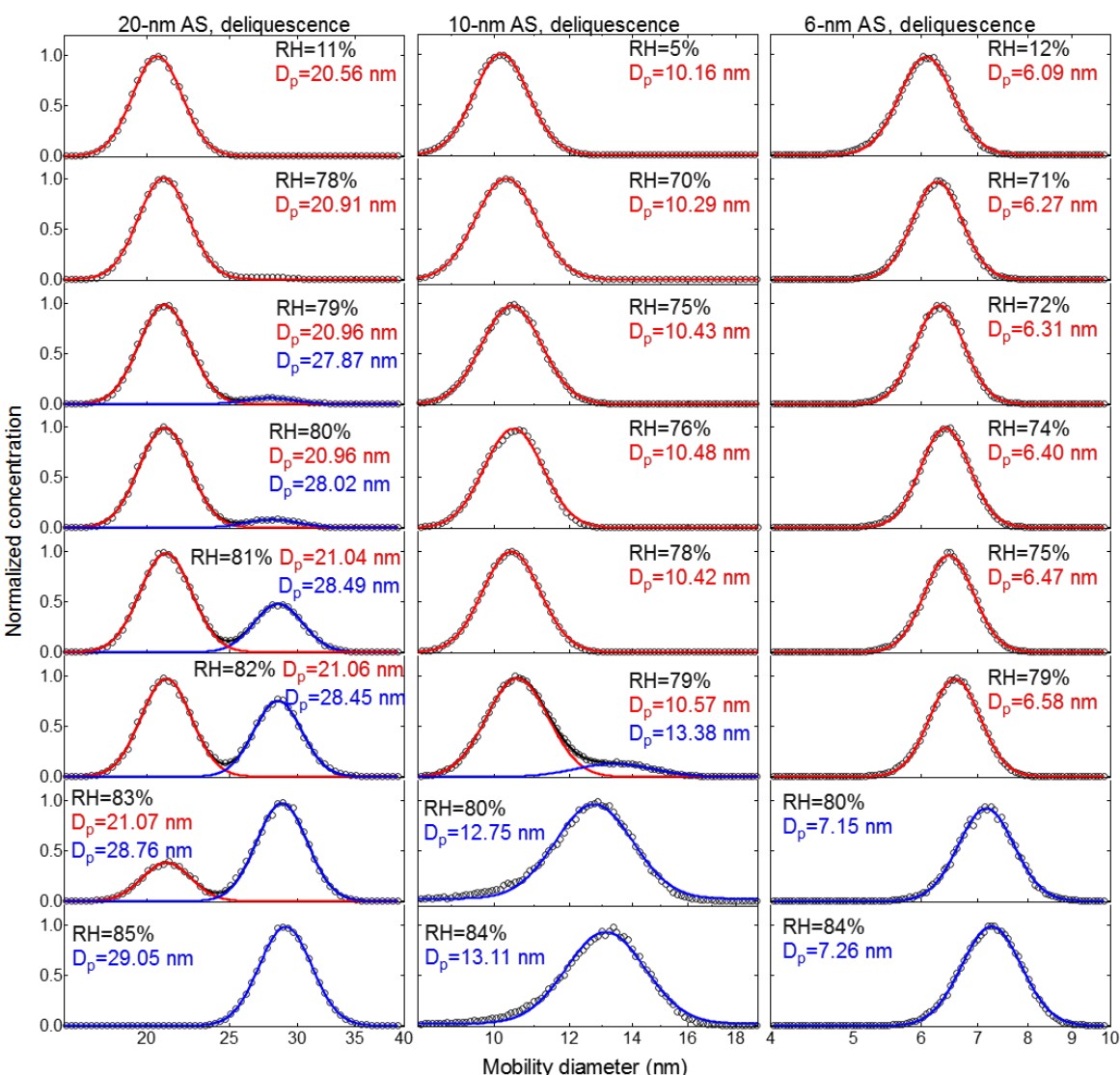


**Figure 5.** Deliquescence-mode measurements of ammonium sulfate (AS) aerosol nanoparticles with dry mobility

diameter from 20-6nm. The measured (black square) and fitted (solid lines) normalized size distribution are shown for

increasing RH. The red and blue lines represent the aerosol nanoparticles in the solid and liquid state, respectively.

The RH history in each measurement is 5% → X%, where X is the RH value given in each panel.

974

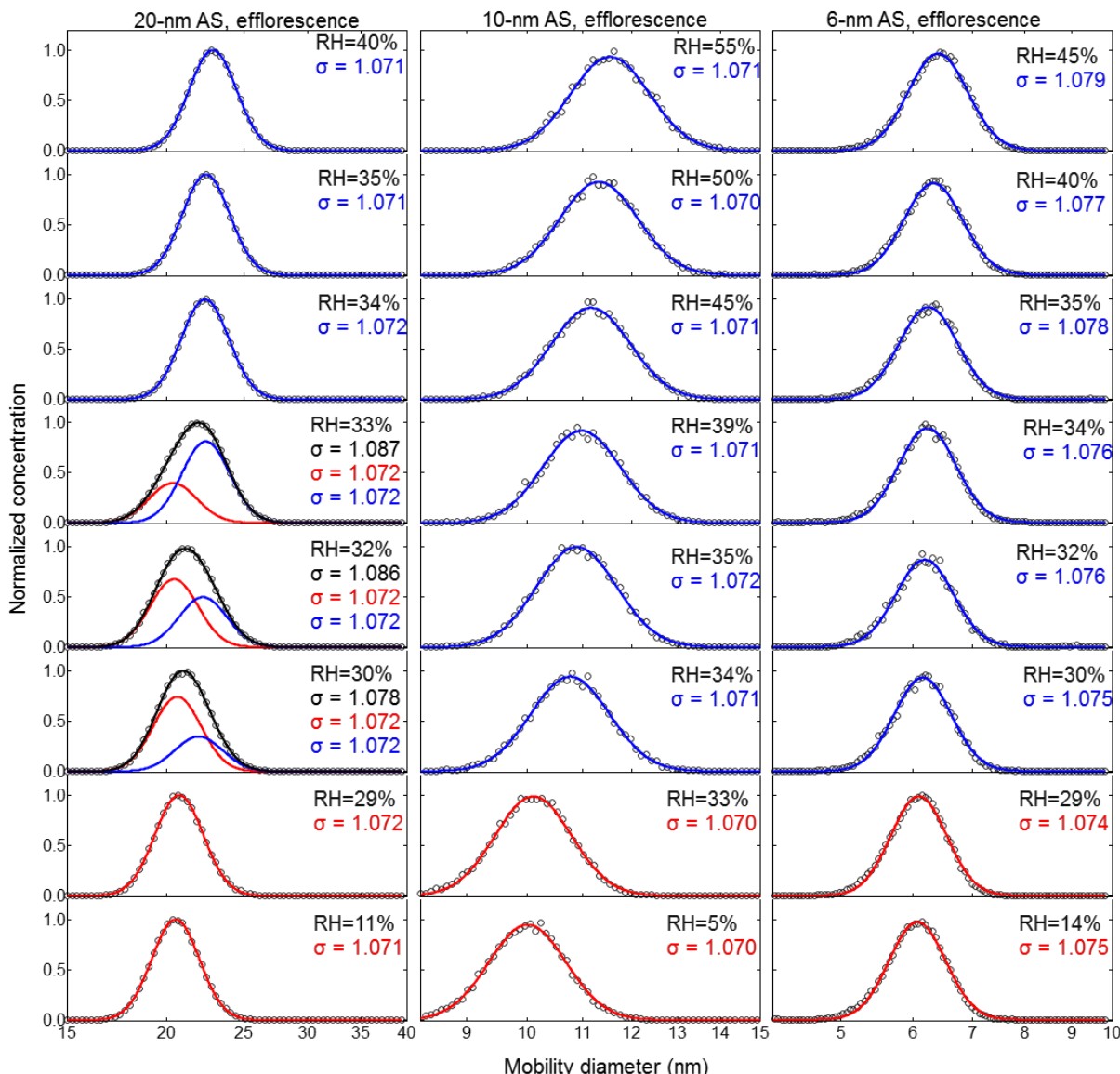

975

**Figure 6.** Efflorescence-mode measurements of ammonium sulfate (AS) aerosol nanoparticles with dry mobility

diameter from 20-6nm. The measured (black circle) and fitted (solid lines) normalized size distribution are shown for

increasing RH. The red and blue lines represent the aerosol nanoparticles in the solid and liquid state, respectively.

The RH history in each measurement is 5%→97%→X%, where X is the RH value given in each panel.




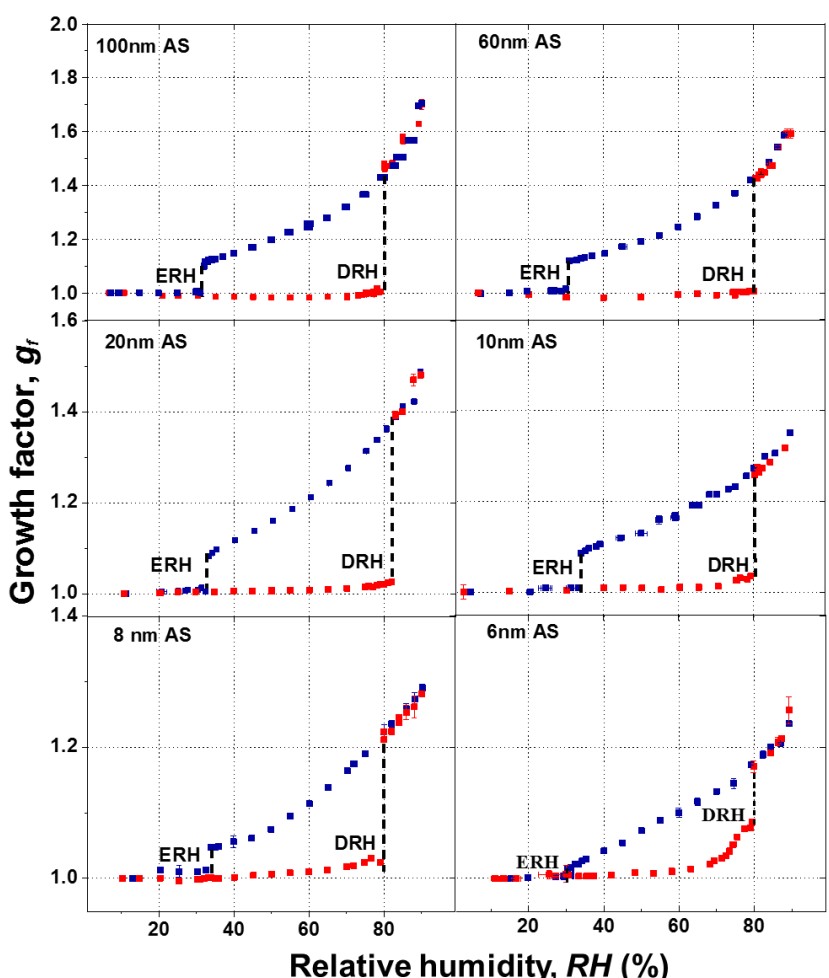


**Figure 7.** Mobility-diameter hygroscopic growth factors ($g_f$) of ammonium sulfate (AS) aerosol nanoparticles with dry

mobility diameter from 6 to 100 nm in the deliquescence mode (red square and error bar) and the efflorescence mode

(royal square and error bar). Deliquescence, and efflorescence relative humidity (DRH&ERH, black dashed line) of

ammonium sulfate (AS) nanoparticles with dry mobility diameter from 6 to 100 nm.



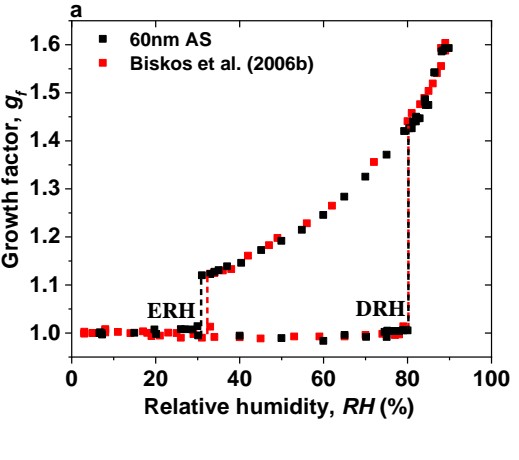
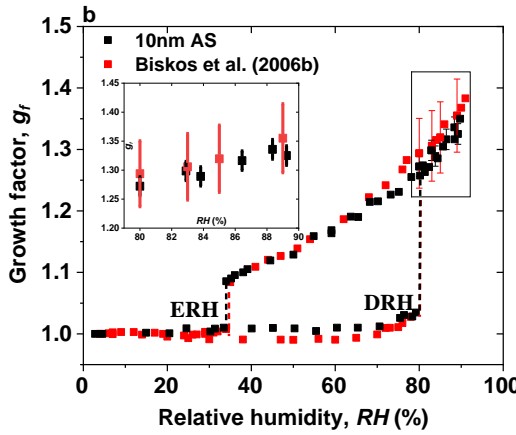


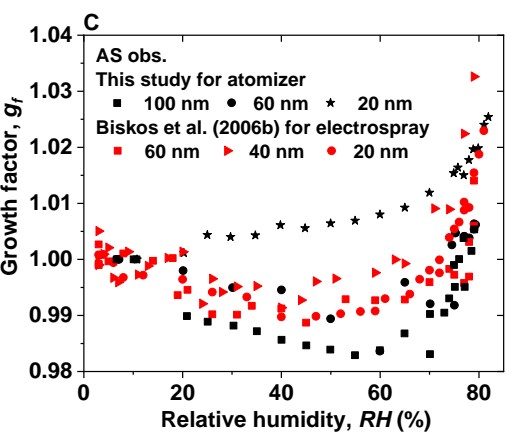
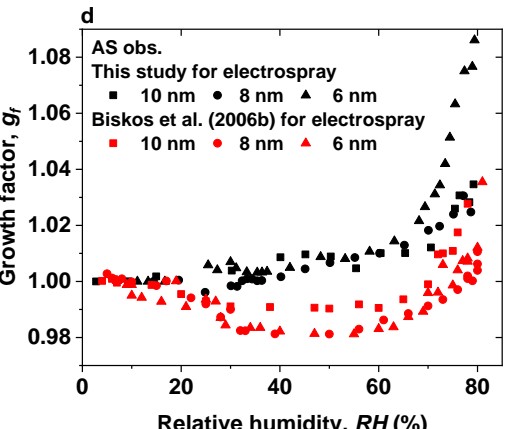

**Figure 8**. **(a-b)** Mobility-diameter hygroscopic growth factors ($g_f$, black squares), deliquescence and efflorescence
relative humidity (DRH&ERH, black dashed lines) of ammonium sulfate (AS) nanoparticles with dry diameter 60 and
10 nm, respectively. Red squares and dashed lines show the respective results from Biskos et al. (2006b), respectively.
Black and red uncertainties of growth factors at certain RH are calculated by $\sqrt{\left(\left(g_f\frac{\sqrt{2}\varepsilon_{Dp}}{D_p}\right)^2 + \left(\varepsilon_{RH}\frac{dg_f}{dRH}\right)^2\right)}$, where
$\varepsilon_{Dp}$, $\varepsilon_{RH}$, and $g_f$ are uncertainty of particle mobility diameter, uncertainty of relative humidity, and growth factor with
respect to RH, respectively (Mochida and Kawamura 2004). **(c-d)** Comparison of growth factors of ammonium sulfate
(AS) nanoparticles with dry diameter range from 6 to 100 nm with Biskos et al. (2006b) prior to deliquescence of
ammonium sulfate nanoparticles.


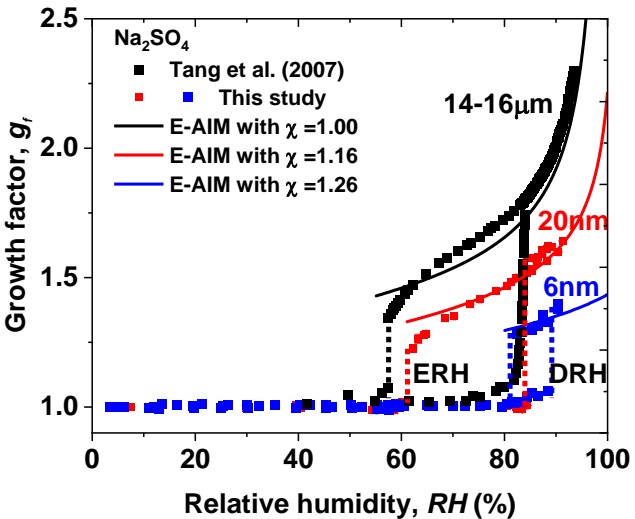


**Figure 9**. Mobility-diameter hygroscopic growth factors ($g_f$), deliquescence and efflorescence relative humidity (DRH&ERH, red and blue dashed lines) of sodium sulfate nanoparticles with dry diameter 20 (red square) and 6 (blue square) nm, respectively. Black squares and dashed lines show the respective results from Tang et al. (2007) with electrodynamic balance (EDB), respectively. In this study, the black, red, and blue curves show E-AIM predictions, including the Kelvin effect and shape factors (χ).






