# Peer review of "Nano-HTDMA for investigating hygroscopic properties of sub-10 nm aerosol"

_Atmospheric Measurement Techniques, 2020_

## Referee Comment (RC1) · Anonymous Referee #1 · 12 May 2020

In the work submitted, Lei et al. presented the design, construction, calibration and validation of a nano-HTDMA apparatus, which can be used to measure hygroscopic growth of aerosol particles down to < 10 nm. The technique they developed is very important, and they also carried out calibration and validation experiments very comprehensively. The paper is also well-written, and I only have a few comments. General comments: Line 827: Compared to "sizing accuracy", "sizing offset" may better describe the actual content of Section 3.1.1. Sections 2.2.1 and Section 3.1.1: I think both sizing accuracy (difference between actual size and the size measured using a

[Figure]

DMA) and sizing offset (i.e. measured difference between the two DMAs) are important for H-TDMA. While sizing offset has been carefully characterized (Section 3.1.1) for particles down to a few nm, not much information has been provided for the sizing accuracy for <100 nm particles. Although experiments to determine size accuracy for <100 nm particles seem to be impossible, as discussed in Section 2.2.1, could the author estimate the sizing accuracy from a theoretical view? Line 300-305: It is interesting to find that sizing offset (<0.9%) is smallest at 8 and 10 nm, smaller than that at smaller diameter (6 nm) and at larger diameter (20 nm or larger). Is there any explanation. Technical comments: Line 57: change "challenge" to "challenging". Line 349-353: I am not sure Wikipedia is a reliable source for physical/chemical constants. I would recommend textbooks/handbooks instead.

---

## Referee Comment (RC2) · Anonymous Referee #2 · 10 Jun 2020

This manuscript "Nano-hygroscopicity tandem differential mobility analyzer (nano-HTDMA) for investigating hygroscopic properties of sub-10 nm aerosol nanoparticles" presents a design of a HTDMA to measure the hygroscopicity particle down to ~6 nm. The performance and the methods to calibrate and validate the setup were also reported. This setup was shown to have low sizing offset (<1.4% for 100 nm particle). High accuracy for flow rates of aerosol and sheath flow (±1%) and high accuracy for voltages applied to DMA (±0.1%) were found to be crucial to achieve the low sizing offset. Also the DMA2 and humidification system were designed to be placed in housing with stable temperate (±0.1K). The RH of sheath flow was set to the same as RH of aerosol flow to prevent the pre-deliquescence. Using this setup, the authors measured the deliquescence and the efflorescence RH as well as the growth factors of ammonium sulfate and sodium sulfate. For ammonium sulfate, no significant size dependence of DRH and ERH was observed while clear size dependence was observed.

Determining the hygroscopicity of nano-particles is important to understand aerosol-water interaction and provides constraints on the chemical composition of nano-particles. This nano-HTDMA has excellent performance and will be useful to measure hygroscopictiy of atmospheric nano-particles. The manuscript is well-written and fit well the scope of AMT. I recommendation its publication in AMT after addressing the following minor comments.

Minor comments

1. What is the smallest particle size that the HTDMA can measure?
   The title "Nano-hygroscopicity tandem differential mobility analyzer (nano-HTDMA) for investigating hygroscopic properties of sub-10 nm aerosol nanoparticles" reads a little redundant for me. In addition, the manuscript discusses many experiments for particle >10 nm. I suggest optimizing the title.
2. Line 428-434, and Fig. 8d, the same method, electrospray was used to generate aerosol <20 nm in this study and the study by Biskos et al. 2006. But the results (growth factors) are still different. Can the authors discuss the difference? Is it possible to generate particles of the same size, i.e. 20 nm with different methods and compare the GF?
3. Fig. 5 and Fig. 7, can the author discuss why the 6 nm AS showed a slight increase with increasing RH.
4. Fig.7, why the DRH for 20 nm AS is different from others (the dashed line)? Also the coloring of efflorescence and deliquescence in this panel contradicts the caption.
5. Line 375-376, "double-mode phenomenon was not observed 375 for 8 and 6 nm ammonium sulfate nanoparticles" . Is this because of the slower mass transfer of water vapor for larger particles?
6. Line 472-474, why does DRH/ERH of sodium sulfate show a clear size dependence while ammonium sulfate does not?

Technical comments

1. Line 347, "excuses air" or "excess air"?
2. Line 427, "continues" should be "continuous".
3. Line 470, "sensitivity" should be "sensitive".
4. Fig. 5 and 6, I suggest explaining the red and blue lines in the captions, although they were explained in the main text.

---

## Author Comment (AC1) · 3 Aug 2020

**Response to comments by anonymous referee #1:**

*In the work submitted, Lei et al. presented the design, construction, calibration and validation of a nano-HTDMA apparatus, which can be used to measure hygroscopic growth of aerosol particles down to < 10 nm. The technique they developed is very important, and they also carried out calibration and validation experiments very comprehensively. The paper is also well-written, and I only have a few comments.*

**Response:** We are grateful to referee #1 for her/his comments and suggestions to improve our manuscript. We have implemented changes based on these comments in the revised manuscript. We repeat the specific points raised by the reviewer in italic font, followed by our response. The pages numbers and lines mentioned are with respect to the Atmospheric Measurement Techniques Discussions (AMTD) version.

**General comments:**

*(1) Compared to "sizing accuracy", "sizing offset" may better describe the actual content of Section 3.1.1. Sections 2.2.1 and Section 3.1.1: I think both sizing accuracy (difference between actual size and the size measured using a DMA) and sizing offset (i.e. measured difference between the two DMAs) are important for H-TDMA. While sizing offset has been carefully characterized (Section 3.1.1) for particles down to a few nm, not much information has been provided for the sizing accuracy for <100 nm particles. Although experiments to determine size accuracy for <100 nm particles seem to be impossible, as discussed in Section 2.2.1, could the author estimate the sizing accuracy from a theoretical view?*

**Response:** Good comment, and thanks. Yes, the reviewer is right, it is not possible to determine size accuracy for < 100 nm particles, and sub-20 nm PSL is even not available. Following the reviewer's suggestion, here we try to estimate the sizing accuracy in this size range through error propagation by using a differential mobility analysis (DMA) transfer function and the uncertainties of its input parameters (Duplissy et al., 2009; Wiedensohler et al., 2012). According to Knutson and Whitby (1975), sizing of DMA transfer function mainly depends on sheath flow rates and high voltage (HV) applied to the DMA as follows:

$$z_p^* = \frac{Q_{sh} \ln \frac{r_2}{r_1}}{2\pi L V} \tag{R1}$$

$$z_p^* = \frac{neC_c}{3\pi\mu d_p^*} \tag{R2}$$

$$d_p^* = \frac{2VLneC_c}{3\mu Q_{sh}ln\frac{r_2}{r_1}} \tag{R3}$$

where $z_p^*$ is the central electrical mobility, $Q_{sh}$ is the sheath flow rate, $V$ is the applied voltage, $L$ is the length of the classification region within the DMA, and $r_1$ and $r_2$ are the inner and outer radii of the DMA annulus, respectively. $n$ is the number of elementary charges of particles. $e$ is the elementary charges. $C_c$ is the slip correction. $\mu$ is the flow viscosity. $d_p^*$ is the mean particle mobility diameter.

According to Eq. (R3) above, we use the following error propagation formula (Eq. (R4)) (Taylor and Taylor, 1997) to calculate the uncertainties in sizing of nanoparticles. In our study, the flow accuracy of mass flow meter (TSI series 4000) is within ±2%. The deviation of voltage applied to the nano-DMAs (0-12500 V, 0-350 V) varies around the set value when test with voltage power supply (HCE 0-12500, HCE 0-350, Fug Electronic) shown in Table R1. Thence, the sizing accuracy is obtained using Eq. (R5) as shown in Table R1.

$$\delta z = \sqrt{\left(\frac{\partial z}{\partial x}\right)^2 (\delta x)^2 + \left(\frac{\partial z}{\partial y}\right)^2 (\delta y)^2} \tag{R4}$$

$$\frac{\delta d}{d} = \sqrt{\left(\frac{\delta V}{V}\right)^2 + \left(\frac{\delta Q_{sh}}{Q_{sh}}\right)^2} \tag{R5}$$

**Table R1 (new Table S5 in revised SI).** The values of size, uncertainty of nano-DMA voltage and sheath flow rates, and calculated size uncertainty.

| Size (nm) | Uncertainty (V, $Q_{sh}$) | Uncertainty (Sizing accuracy) |
|---|---|---|
| 100 | 2648.2±0.02592 V, 10±0.02 L/min | 0.2000% |
| 60 | 1063.0±0.02686 V, 10±0.02 L/min | 0.2000% |
| 20 | 131.1±0.01519 V, 10±0.02L/min | 0.2003% |
| 10 | 33.7±0.02435 V, 10±0.02 L/min | 0.2127% |
| 8 | 21.6±0.03725 V, 10±0.02 L/min | 0.2641% |
| 6 | 12.2±0.06920 V, 10±0.02 L/min | 0.6014% |

**Related additions and changes included in the revised manuscript:**

**Page 13 line 299, we add:** "Sizing accuracy of sub-100 nm aerosol nanoparticles, as discussed in Sec. 2.2.1, is even impossible to verify using PSL nanoparticles. Duplissy et al. (2009) and Wiedensohler et al. (2012) suggested that sizing accuracy of sub-100 nm nanoparticles could be test by a DMA transfer function. The theoretical DMA transfer function (see SI. Eq. (S2-S4)) was proposed by Knutson and Whitby (1975) and they noted that sizing is crucially dependent on flow rates and high voltage (HV) applied to the DMA. Thence, for nanoparticles with diameter smaller than 100 nm, in our study, the flow accuracy of mass flow meter (TSI series 4000) is within ±2%. The deviation of voltage applied to the nano-DMAs (0-12500 V, 0-350 V) varies around the set value when test with voltage power supply (HCE 0-12500, HCE 0-350, Fug Electronic) shown in Table S5. According to the error propagation formula (see SI. Eq. (S5)) (Taylor and Taylor, 1997). The calculated uncertainty in sizing of 6-100 nm nanoparticles increases as size decreases, which is roughly consistent with measured sizing accuracy and sizing offset of two nano-DMAs (see SI. Table S5). However, the calculated sizing accuracy is smaller than measured sizing accuracy. This suggested uncertainties of slip correction, DMA dimensions (inner and outer radius, length), temperature, pressure, and viscosity of air could affect the sizing accuracy according to Eq. (S4) (Kinney et al., 1991). Besides DMA transform function, Wiedensohler et al. (2012) suggested that the possible sources of uncertainty of sizing are particle losses, the size- and material-dependent CPC counting efficiency, which results in a bigger sizing deviation of nanoparticle during the measurements compared to the estimated sizing accuracy according to theory."

**Related additions included in the supplementary information:**

**Line 156, we add:**

**S2. Calculation of sizing accuracy of sub-100 nanoparticles**

Knutson and Whitby (1975) proposed the following theoretical differential mobility analyzer (DMA) transfer function and showed that sizing is crucially dependent on sheath flow rates and high voltage (HV) applied to the DMA.

$$z_p^* = \frac{Q_{sh} ln\frac{r_2}{r_1}}{2\pi LV} \tag{S2}$$

$$z_p^* = \frac{neC_c}{3\pi\mu d_p^*} \tag{S3}$$

$$d_p^* = \frac{2VLneC_c}{3\mu Q_{sh} ln\frac{r_2}{r_1}} \tag{S4}$$

where $z_p^*$ is the central electrical mobility, $Q_{sh}$ is the sheath flow rate, $V$ is the applied voltage, $L$ is the length of the classification region within the DMA, and $r_1$ and $r_2$ are the inner and outer radii of the DMA annulus, respectively. $n$ is the number of elementary charges of particles. $e$ is the elementary charges. $C_c$ is the slip correction. $\mu$ is the flow viscosity. $d_p^*$ is the mean particle mobility diameter.

According to Eq. (S4) above, we use the following error propagation formula ((Taylor and Taylor, 1997) to calculate the uncertainties in sizing of nanoparticles. In our study, the flow accuracy of mass flow meter (TSI series 4000) is within ±2%. The deviation of voltage applied to the nano-DMAs (0-12500 V, 0-350 V) varies around the set value when test with voltage power supply (HCE 0-12500, HCE 0-350, Fug Electronic) shown in Table S5. Thence, the uncertainties in sizing of nanoparticles are obtained based on the following Eq. (S5) as shown in Table S5.

$$\frac{\delta d}{d} = \sqrt{\left(\frac{\delta V}{V}\right)^2 + \left(\frac{\delta Q_{sh}}{Q_{sh}}\right)^2} \tag{S5}$$

*(2) Line 300-305: It is interesting to find that sizing offset (<0.9%) is smallest at 8 and 10 nm, smaller than that at smaller diameter (6 nm) and at larger diameter (20 nm or larger). Is there any explanation.*

**Response:** Thanks for the comment. Uncertainties in the sheath flow rates and nano-DMA voltages will increase as size decreases, which results in a larger size offset of 6-nm nanoparticles compared with other sizes. However, we observed that the peak diameter of number size distribution of the generated pure water is ~20-30 nm (Figure R1), which is more likely due to presence of impurities in the water. This interferes the accurate measurement of 20-nm nanoparticles.

[Figure]

**Figure R1.** Number concentration scanned for water nanoparticles by the nano-DMA2 at RH below 5 % at 298 K.

**Page 13 line 305, we add**: "Uncertainties in the sheath flow rates and nan-DMA voltages will increase as size decreases, which results in a larger sizing offset of 6-nm nanoparticles compared with other sizes."

*Technical comments:*

*(1) Line 57: change "challenge" to "challenging".*
**Response:** Many thanks. We have revised in the following sentence and now they read as:
**Page 3 line 55-57:** "In addition, by knowing the hygroscopicity of newly formed nanoparticle, one can infer the involving chemical species (e.g., organic ratio) in particle formation and initial growth (Wang et al., 2010), which is otherwise difficult and highly challenging to measure directly (Wang et al., 2010; Ehn et al., 2014)."

*(2) Line 349-353: I am not sure Wikipedia is a reliable source for physical/chemical constants. I would recommend textbooks/handbooks instead.*
**Response:** Thanks for your suggestions. We have cited Atkins et al. (2006) in the following sentence:
**Page 15 line 349-353:** "It may due to the heat produced from the inner electrode of nano-DMA2, which we estimated to be ~0.08 W ($Q = mdTC_{p,}$) by considering the density and heating capacity of air, and aerosol and sheath air flow rate ($\rho$=1.2041kg/m$^3$; $C_p$=1.859kJ/kg°C) (Atkins et al., 2006)."

**Reference:**

Atkins, P., De Paula, J., and Walters, V.: Physical Chemistry, W. H. Freeman, 2006.

Kinney, P. D., Pui, D. Y. H., Mullholland, G. W. & Bryner, N. P. Use of the Electrostatic Classification Method to Size 0.1 μm SRM Particles—A Feasibility Study. Journal of Research of the National Institute of Standards and Technology, 96, 147, 1991.

Knutson, E. O. and Whitby, K. T.: Aerosol classification by electric mobility: apparatus, theory, and applications, Journal of Aerosol Science, 6, 443-451, 1975.

Taylor, J. R. and Taylor, S. L. L. J. R.: Introduction To Error Analysis: The Study of Uncertainties in Physical Measurements, University Science Books, 1997.

Wiedensohler, A., Birmili, W., Nowak, A., Sonntag, A., Weinhold, K., Merkel, M., Wehner, B., Tuch, T., Pfeifer, S., Fiebig, M., Fjäraa, A. M., Asmi, E., Sellegri, K., Depuy, R., Venzac, H., Villani, P., Laj, P., Aalto, P., Ogren, J. A., Swietlicki, E., Williams, P., Roldin, P., Quincey, P., Hüglin, C., Fierz-Schmidhauser, R., Gysel, M., Weingartner, E., Riccobono, F., Santos, S., Grüning, C., Faloon, K., Beddows, D., Harrison, R., Monahan, C., Jennings, S. G., O'Dowd, C. D., Marinoni, A., Horn, H. G., Keck, L., Jiang, J., Scheckman, J., McMurry, P. H., Deng, Z., Zhao, C. S., Moerman, M., Henzing, B., de Leeuw, G., Löschau, G., and Bastian, S.: Mobility particle size spectrometers: harmonization of technical standards and data structure to facilitate high quality long-term observations of atmospheric particle number size distributions, Atmos. Meas. Tech., 5, 657-685, 2012.

---

## Author Comment (AC2) · 3 Aug 2020

Nano-hygroscopicity tandem differential mobility analyzer (nano-HTDMA) for investigating hygroscopic properties of sub-10 nm aerosol nanoparticles
journal article response
Atmospheric Measurement Techniques
en

*Response to comments by anonymous referee #2:*

*This manuscript "Nano-hygroscopicity tandem differential mobility analyzer (nano-HTDMA) for investigating hygroscopic properties of sub-10 nm aerosol nanoparticles" presents a design of a HTDMA to measure the hygroscopicity particle down to ~6 nm. The performance and the methods to calibrate and validate the setup were also reported. This setup was shown to have low sizing offset (<1.4% for 100 nm particle). High accuracy for flow rates of aerosol and sheath flow (±1%) and high accuracy for voltages applied to DMA (±0.1%) were found to be crucial to achieve the low sizing offset. Also the DMA2 and humidification system were designed to be placed in housing with stable temperate (±0.1K). The RH of sheath flow was set to the same as RH of aerosol flow to prevent the pre-deliquescence. Using this setup, the authors measured the deliquescence and the efflorescence RH as well as the growth factors of ammonium sulfate and sodium sulfate. For ammonium sulfate, no significant size dependence of DRH and ERH was observed while clear size dependence was observed. Determining the hygroscopicity of nano-particles is important to understand aerosol-water interaction and provides constraints on the chemical composition of nano-particles. This nano-HTDMA has excellent performance and will be useful to measure hygroscopictiy of atmospheric nano-particles. The manuscript is well-written and fit well the scope of AMT. I recommendation its publication in AMT after addressing the following minor comments.*

**Response:** We are grateful to referee #2 for the comments and the constructive suggestions. We address in the following the comments and suggestions by referee #2 and provide improvements based on these clarify the questioned issues in the revised manuscript. The pages numbers and lines mentioned are with respect to the Atmospheric Measurement Techniques Discussions (AMTD) version.

*Minor comments:*

*(1) What is the smallest particle size that the HTDMA can measure?*

*The title "Nano-hygroscopicity tandem differential mobility analyzer (nano-HTDMA) for investigating hygroscopic properties of sub-10 nm aerosol nanoparticles" reads a little redundant for me. In addition, the manuscript discusses many experiments for particle >10 nm. I suggest optimizing the title.*

**Response:** Thanks for the comment. At the moment, the smallest size that we can measure is 6 nm. The main purpose of the instrument development is to have a device that is able to measure hygroscopic growth of sub-10 nm nanoparticles. We discussed that the results of 20 nm and 100 nm is to compare with literature studies, which are the most abundant (especially for 100 nm) and also is to demonstrate the differences between measuring hygroscopic growth of sub-10 nm nanoparticles and larger ones. Following the suggestion, we revised the tittle as "Nano-HTDMA for investigating hygroscopic properties of sub-10 nm aerosol nanoparticles".

**Related additions and changes included in the revised manuscript:**

**Page 1 line 1-2:** "Nano-HTDMA for investigating for hygroscopic properties of sub-10 nm aerosol nanoparticles".

*(2) Line 428-434, and Fig. 8d, the same method, electrospray was used to generate aerosol <20 nm in this study and the study by Biskos et al. 2006. But the results (growth factors) are still different. Can the authors discuss the difference? Is it possible to generate particles of the same size, i.e. 20 nm with different methods and compare the GF?*

**Response:** Thanks for the comment.

The morphology of particles may affect their hygroscopic behavior (Mikhailov et al., 2004, 2009). Iskandar et al. (2003) and Wang et al. (2019) show that the morphology of the aerosol particles mainly depends on initial properties of droplets (e.g., chemical composition and solution concentration) and drying process. In Table 1, we compared the generation conditions with Biskos et al. 2006b for 6-10 nm ammonium sulfate nanoparticles using an electrospray. Different from generation conditions in Biskos et al. (2006b) for 6-10 nm ammonium sulfate nanoparticles, in our study, in order to minimize the multiple charged nanoparticles, three different concentrations are used so that the sizes (e.g., 6, 8, 10 nm) selected by the nano-DMA1 were always slight larger than peak of the number size distribution of the generated nanoparticles by the electrospray. This is to ensure that we could have as many as nanoparticles as possible to compensate the strong nanoparticle losses in the nano-HTDMA system. Besides different generation conditions shown in Table R1, the drying rate is mainly dependent on drying flow rate in the HTDMA system (Wang et al., 2019). The RH of dried ammonium sulfate aerosol nanoparticles varies due to the different aerosol/sheath flow rates employed in Biskos et al. (2006b) and this study, respectively. These

differences may lead to the small difference in growth factor of ammonium sulfate nanoparticles prior to the deliquescence of ammonium sulfate.

**Page 19 line 428, we add**: "Due to the water adsorption on the surface of nanoparticles, the morphology of particles may change and further affect their hygroscopic behavior (Mikhailov et al., 2004, 2009). Iskandar et al. (2003) and Wang et al. (2019) show that the morphology of the aerosol particles mainly depends on generation conditions (e.g., chemical composition, solution concentration) and drying process."

**Table R1**. Comparison of generation of ammonium sulfate (AS) nanoparticles with diameter from 6-10 nm with Biskos et al. (2006b) using an electrospray

| Generation of 6-10 nm AS nanoparticles | AS concentration (mM) | Size of capillary | Flow rates | RH of generated AS nanoparticles |
|---|---|---|---|---|
| Biskos et al. (2006b) | 10 | 40 μm | 2 l/min dry air | 0.1% |
| This study | 1, 5, 20 | 20 μm | 1 l/min dry $N_2$ | 2% |

Following reviewer's suggestion, we used an electrospray and an atomizer to generate 20-nm ammonium sulfate aerosol nanoparticles, respectively. We then compared their hygroscopic growth factors prior to deliquescence. Figure R1a shows a ~ 0.1 higher growth factor of 20-nm ammonium sulfate generated by an electrospray than that using an atomizer in the RH range from 55% to 82%. Also, a slight difference in hygroscopic growth factor of 20-nm NaCl aerosol nanoparticles is observed in previous study using the different generation methods (Biskos et al., 2006a). Figure R1b shows the results of 20-nm sodium chloride nanoparticles using an electrospray and a vaporization-condensation method (Biskos et al., 2006a), respectively. There is a slight difference in the growth factor of 20-nm sodium chloride at RH between 20% and 60% using the different generation methods.

**Page 19 line 432, we add**: "Different from generation conditions in Biskos et al. (2006b) for 6-10 nm ammonium sulfate nanoparticles, in our study, in order to minimize the multiple charged nanoparticles, three different concentrations are used so that the sizes (e.g., 6, 8, 10 nm) selected by the nano-DMA1 were always slight larger than peak of the number size distribution of the generated nanoparticles by the electrospray. This is to ensure that we could have as many as nanoparticles as possible to compensate the strong nanoparticle losses in the nano-HTDMA system.

Also, we used an electrospray and an atomizer to generate 20-nm ammonium sulfate, respectively, and then compared their hygroscopic growth factors prior to deliquescence. Figure S13a shows a ~ 0.1 higher growth factor of 20-nm ammonium sulfate generated by an electrospray than that using an atomizer in the RH range from 55% to 82%, which is similar to a slight difference in hygroscopic growth factor of 20-nm NaCl aerosol nanoparticles as observed in Fig S13b in Biskos et al. (2006a) using the different generation methods. Besides different generation conditions, the drying rate is mainly dependent on drying flow rate in the HTDMA system (Wang et al., 2019). The RH of dried ammonium sulfate aerosol nanoparticles varies due to the different aerosol/sheath flow rates employed in Biskos et al. (2006b) and this study, respectively."

[Figure]

**Figure R1 (new Figure S13 in revised SI)**. Hygroscopic growth factors of 20-nm **(a)** ammonium sulfate (AS) nanoparticles and **(b)** sodium chloride (NaCl) nanoparticles from Biskos et al. (2006a) using the different generation methods prior to deliquescence.

*(3) Fig. 5 and Fig. 7, can the author discuss why the 6 nm AS showed a slight increase with increasing RH.*

**Response:** Thanks for the comments. Yes, a slight increase in hygroscopic growth factor of 6-nm ammonium sulfate nanoparticles was observed in the RH range from 65 to 79% RH before deliquesces. This is attributed to water adsorption onto the surfaces of these nanoparticles. It seems that there is more water adsorption onto the small nanoparticles than that of large nanoparticles. Similar phenomenon has also observed by Hämeri et al. (2000, 2001), Romakkaniemi et al. (2001), Biskos et al. (2006a, b, 2007), and Giamarelou et al. (2018). The reason for such enhanced adsorption is still to be investigated.

**Page 19 line 428, we added:** "For example, a slight increase in hygroscopic growth factor of 6-nm ammonium sulfate nanoparticles is observed in the RH range from 65 to 79% RH before deliquescence. This is attributed to water adsorption onto the surfaces of these nanoparticles. It seems that there is more water adsorption onto small nanoparticles than that of large nanoparticles. Similar phenomenon has also observed by Hämeri et al. (2000, 2001), Romakkaniemi et al. (2001), Biskos et al. (2006a, b, 2007), and Giamarelou et al. (2018). The reason for such enhanced adsorption is still to be investigated."

*(4) Fig.7, why the DRH for 20 nm AS is different from others (the dashed line)? Also the coloring of efflorescence and deliquescence in this panel contradicts the caption.*

**Response:** Thanks for the comment. The DRH of 20-nm ammonium sulfate is slightly different from that at other sizes. Also, the similar phenomenon was observed for 20-nm ammonium sulfate nanoparticles from Biskos et al. (2006b) shown in Fig. R2, which shows in agreement with our study. To my knowledge, we observed that the peak diameter of number size distribution of pure water is ~20-30 nm (Figure S2a), which is more likely due to presence of impurities in the water. This interferes the accurate measurement of 20-nm nanoparticles.

[Figure]

**Figure R2**. Comparison of the hygroscopic behavior of 20-nm ammonium sulfate (AS) with Biskos et al. (2006b).

**Page 45 line 917, we revised the color of 20-nm ammonium sulfate in both deliquescence and efflorescence measurement modes and made the color consistent with citation:**

[Figure]

**Figure 7.** Mobility-diameter hygroscopic growth factors ($g_f$) of ammonium sulfate (AS) aerosol nanoparticles with dry mobility diameter from 6 to 100 nm in the deliquescence mode (red square and error bar) and the efflorescence mode (royal square and error bar). Deliquescence, and efflorescence relative humidity (DRH&ERH, black dashed line) of ammonium sulfate (AS) nanoparticles with dry mobility diameter from 6 to 100 nm.

*(5) Line 375-376, "double-mode phenomenon was not observed 375 for 8 and 6 nm ammonium sulfate nanoparticles". Is this because of the slower mass transfer of water vapor for larger particles?*

**Response:** Many thanks. No, this is not because of the slower mass transfer of water vapor for larger sulfate nanoparticles. Double-mode phenomenon was observed for 99-nm sodium chloride in Mikhailov et al. (2004) and for 10-nm ammonium sulfate and sodium chloride in Biskos et al. (2006b, 2007) in the deliquescence measurement mode, respectively. They attributed this to the co-existence of solid and liquid phase of aerosol nanoparticles due to the slight inhomogeneity of RH within nano-DMA2. Bezantakos et al. (2016) have shown the difference of RH for sheath flow and aerosol flow upstream of DMA2 and temperature gradient within DMA2 can result in RH non-uniformities within DMA2. In our study, we also observed this double-mode phenomenon for ammonium sulfate nanoparticles with diameters (e.g., 100, 60, 20, 10 nm) but not for 8 and 6 nm ammonium sulfate nanoparticles. Because this phenomenon is an essentially stochastic process.

*(6) Line 472-474, why does DRH/ERH of sodium sulfate show a clear size dependence while ammonium sulfate does not?*

**Response:** Many thanks.

No significant size effect on the DRH and ERH of 6-100 nm ammonium sulfate nanoparticles is mainly due to the strong non-ideality of aqueous ammonium sulfate solution property (Cheng et al. 2015). The different responses of DRH and ERH of sodium chloride and ammonium sulfate on changing particle size have been theoretically studied and explained by Cheng et al. (2015). They show that, as presented in Fig. R3, the main reason for this phenomenon is that the stronger increase in solute molality of ammonium sulfate is required for the same change of solute activity than that of sodium chloride, although the different solute-liquid surface tension may also play a role. The phase transition concentration (deliquescence and crystallization concentration) of ammonium sulfate is thus more sensitive to the size range from 6 to 100 nm compared to that of sodium chloride. This leads to the almost unchanged DRH and ERH of ammonium sulfate nanoparticles (Cheng et al., 2015).

[Figure]

**Figure R3**. Saturation ratio of solute activity ($a_s^*/a_{s,bulk}^*$) as a function of molality $b$ for ammonium sulfate (AS) and sodium chloride (NaCl). Reprinted with permission by Cheng et al. (2015).

For the size dependence of phase transition of sodium sulfate, there is a clear size effect on DRH and ERH similar to that of sodium chloride but different from that of ammonium sulfate in the size range from 6 to 20 nm, suggesting that non-ideality of solution property is close to that of sodium chloride but weaker than that of ammonium sulfate.

**Page 20 line 466-477, we revised:** "The strong size-effect on the DRH and ERH of sodium chloride and on hygroscopic growth factors of ammonium sulfate have been observed by Biskos et al. (2006a, b, 2007) and theoretically studied and explained by Cheng et al. (2015). Owning to the strong non-ideality of aqueous ammonium sulfate solution, the phase transition concentration (deliquescence and crystallization concentration) of ammonium sulfate is much more sensitivity to the size changes from 60 nm to 6 nm than that of sodium chloride, leading to the almost unchanged DRH and ERH of ammonium sulfate nanoparticles (Cheng et al., 2015). Compared the three compounds, the size-dependent hygroscopicity of sodium sulfate nanoparticles from 20 nm to 6 nm is similar to that of sodium chloride, but different to that of ammonium sulfate, where no significant change in DRH and ERH was observed. However, in this size range, the increase of the ERH and the decrease of growth factor upon decreasing size seems to be stronger for sodium sulfate than sodium chloride, although no significant change in DRH was observed from micrometer size particles down to 20 nm." **as**

"Different from ammonium sulfate, of which no significant size effect on the DRH and ERH was observed, there is a strong size effect of DRH and ERH of sodium sulfate with diameter down to 6 nm. The different responses of DRH and ERH of sodium chloride and ammonium sulfate on changing particle size have been theoretically studied and explained by Cheng et al. (2015). They explained that the main reason for this phenomenon is that the stronger increase in solute molality of ammonium sulfate is required for the same change of solute activity than that of sodium chloride, although the different solute-liquid surface tension may also play a role. The phase transition concentration (deliquescence and crystallization concentration) of ammonium sulfate is thus more sensitive to the size range from 6 to 100 nm compared to that of sodium chloride. This leads to the almost unchanged DRH and ERH of ammonium sulfate nanoparticles (Cheng et al., 2015). For the size dependence of phase transition of sodium sulfate, the size effect on DRH and ERH is similar to that of sodium chloride but different from that of ammonium sulfate in the size range from 6 to 20 nm, suggesting that non-ideality of solution property is close to that of sodium chloride but weaker than that of ammonium sulfate."

***Technical comments:***

*(1). Line 347, "excuses air" or "excess air"?*

**Response:** Many thanks. We have carefully checked and revised the whole of manuscript and supplement information, including grammar, wording, and sentence structure.

**Page 15 line 347-349:** "we monitored that the sheath flow temperature at the inlet of nano-DMA2 is slightly lower (less than ~0.2 K) than that at the outlet, i.e., the RHs at the inlet of nano-DMA2 is slightly higher (~ 1%) than the RH of the excess air at the outlet."

*(2). Line 427, "continues" should be "continuous".*

**Response:** Many thanks. We have revised in the following sentence and now they read as:

**Page 18 line 426-428:** "There seems to be continuous water adsorption and the adsorbed water layers (Romakkaniemi et al., 2001) become significantly thicker when RH closer to the DRH (i.e, RH > 70%)."

*(3). Line 470, "sensitivity" should be "sensitive".*

**Response:** Many thanks. We have revised in the following sentence and now they read as:

**Page 20 line 468-472:** "Owning to the strong non-ideality of aqueous ammonium sulfate solution, the phase transition concentration (deliquescence and crystallization concentration) of ammonium sulfate is much more sensitive to the size changes from 60 nm to 6 nm than that of sodium chloride, leading to the almost unchanged DRH and ERH of ammonium sulfate nanoparticles (Cheng et al., 2015)."

*(4). Fig. 5 and 6, I suggest explaining the red and blue lines in the captions, although they were explained in the main text.*

**Response:** Many thanks. We add explanations of the red and blue lines in the all captions in the manuscript and supplement information, respectively.

**Page 44 line 907-909:** "**Figure 5.** Deliquescence-mode measurements of ammonium sulfate (AS) aerosol nanoparticles with dry mobility diameter from 20-6nm. The measured (black square) and fitted (solid lines) normalized size distribution are shown for increasing RH. The red and blue lines represent the aerosol nanoparticles in the solid and liquid state, respectively. The RH history in each measurement is 5% → X%, where X is the RH value given in each panel."

**Page 45 line 912-914:** "**Figure 6.** Efflorescence-mode measurements of ammonium sulfate (AS) aerosol nanoparticles with dry mobility diameter from 20-6nm. The measured (black circle) and fitted (solid lines) normalized size distribution are shown for increasing RH. The red and blue lines represent the aerosol nanoparticles in the solid and liquid state, respectively. The RH history in each measurement is 5%→97%→X%, where X is the RH value given in each panel."

**Related additions and changes included in the revised supplement information:**

**Line 44-47:** "**Figure S4.** Deliquescence-mode **(a)** and efflorescence-mode **(b)** of 100-nm ammonium sulfate (AS) aerosol nanoparticles. The measured (black square) and fitted (solid lines) normalized size distribution are shown for increasing RH (5%→X%, where X is the RH value given in each panel) and decreasing RH (5%→97%→X%, where X is the RH value given in each panel), respectively. The red and blue lines represent the aerosol nanoparticles in the solid and liquid state, respectively."

**Line 52-55:** "**Figure S5.** Deliquescence-mode **(a)** and efflorescence-mode **(b)** of 60-nm ammonium sulfate (AS) aerosol nanoparticles. The measured (black square) and fitted (solid lines) normalized size distribution are shown for

increasing RH (5%→X%, where X is the RH value given in each panel) and decreasing RH (5%→97%→X%, where X is the RH value given in each panel), respectively. The red and blue lines represent the aerosol nanoparticles in the solid and liquid state, respectively."

**Line 59-62: "Figure S6.** Deliquescence-mode **(a)** and efflorescence-mode **(b)** of 8-nm ammonium sulfate (AS) aerosol nanoparticles. The measured (black square) and fitted (solid lines, single-mode log-normal fit) normalized size distribution are shown for increasing RH (5%→X%, where X is the RH value given in each panel) and decreasing RH (5%→97%→X%, where X is the RH value given in each panel), respectively. The red and blue lines represent the aerosol nanoparticles in the solid and liquid state, respectively."

**Line 95-100: "Figure S9.** Deliquescence-mode **(a)** and efflorescence-mode **(b)** of 20-nm sodium sulfate aerosol nanoparticles. The measured (black square) and fitted (solid lines) normalized size distribution are shown for increasing RH (5%→X%, where X is the RH value given in each panel) and decreasing RH (5%→97%→X%, where X is the RH value given in each panel), respectively. Red/blue solid line is fitted by a single-mode log-normal fit. Red, blue, and black lines are fitted by a double-mode log-normal fit. The red and blue lines represent the aerosol nanoparticles in the solid and liquid state, respectively. The voltage applied to the nano-DMAs (0-12500 V) is kept within ±1% around the set value shown in the voltage meter."

**Line 104-109: "Figure S10.** Deliquescence-mode **(a)** and efflorescence-mode **(b)** of 6-nm sodium sulfate aerosol nanoparticles. The measured (black square) and fitted (solid lines) normalized size distribution are shown for increasing RH (5%→X%, where X is the RH value given in each panel) and decreasing RH (5%→97%→X%, where X is the RH value given in each panel), respectively. Red/blue solid line is fitted by a single-mode log-normal fit. Red, blue, and black lines are fitted by a double-mode log-normal fit. The red and blue lines represent the aerosol nanoparticles in the solid and liquid state, respectively. The voltage applied to the nano-DMAs (0-350 V) is kept within ±1% around the set value shown in the voltage meter."

**Reference**

Bezantakos, S., Huang, L., Barmpounis, K., Martin, S. T., and Biskos, G.: Relative humidity non-uniformities in Hygroscopic Tandem Differential Mobility Analyzer measurements, Journal of Aerosol Science, 101, 1-9, 2016.

Biskos, G., Malinowski, A., Russell, L. M., Buseck, P. R., and Martin, S. T.: Nanosize Effect on the Deliquescence and the Efflorescence of Sodium Chloride Particles, Aerosol Science and Technology, 40, 97-106, 2006a.

Biskos, G., Paulsen, D., Russell, L. M., Buseck, P. R., and Martin, S. T.: Prompt deliquescence and efflorescence of aerosol nanoparticles, Atmospheric Chemistry and Physics, 6, 4633-4642, 2006b.

Biskos, G., Russell, L. M., Buseck, P. R., and Martin, S. T.: Nanosize effect on the hygroscopic growth factor of aerosol particles, Geophysical Research Letters, 33, 2007.

Cheng, Y., Su, H., Koop, T., Mikhailov, E., and Poschl, U.: Size dependence of phase transitions in aerosol nanoparticles, Nature communications, 6, 5923, 2015.

Giamarelou, M., Smith, M., Papapanagiotou, E., Martin, S. T., and Biskos, G.: Hygroscopic properties of potassium-halide nanoparticles, Aerosol Science and Technology, 52, 536-545, 2018.

Hämeri, K., Laaksonen, A., Väkevä, M., and Suni, T.: Hygroscopic growth of ultrafine sodium chloride particles, Journal of Geophysical Research: Atmospheres, 106, 20749-20757, 2001.

Hämeri, K., Väkevä, M., Hansson, H.-C., and Laaksonen, A.: Hygroscopic growth of ultrafine ammonium sulfate aerosol measured using an ultrafine tandem differential mobility analyzer, Journal of Geophysical Research: Atmospheres, 105, 22231-22242, 2000.

Iskandar, F., Gradon, L., and Okuyama, K.: Control of the morphology of nanostructured particles prepared by the spray drying of a nanoparticle sol, Journal of Colloid and Interface Science, 265, 296-303, 2003.

Mikhailov, E., Vlasenko, S., Martin, S. T., Koop, T., and Poschl, U.: Amorphous and crystalline aerosol particles interacting with water vapor: conceptual framework and experimental evidence for restructuring, phase transitions and kinetic limitations, Atmospheric Chemistry and Physics, 9, 9491-9522, 2009.

Mikhailov, E., Vlasenko, S., Niessner, R., and Poschl, U.: Interaction of aerosol particles composed of protein and salts with water vapor: hygroscopic growth and microstructural rearrangement, Atmospheric Chemistry and Physics, 4, 323-350, 2004.

Romakkaniemi, S., Hämeri, K., Väkevä, M., and Laaksonen, A.: Adsorption of Water on 8−15 nm NaCl and $(NH_4)_2SO_4$ Aerosols Measured Using an Ultrafine Tandem Differential Mobility Analyzer, The Journal of Physical Chemistry A, 105, 8183-8188, 2001.

Wang, X., Ma, N., Lei, T., Größ, J., Li, G., Liu, F., Meusel, H., Mikhailov, E., Wiedensohler, A., and Su, H.: Effective density and hygroscopicity of protein particles generated with spray-drying process, Journal of Aerosol Science, 137, 105441, 2019.

---

## Author Response (AR1)

*Response to comments by anonymous referee #1:*

*In the work submitted, Lei et al. presented the design, construction, calibration and validation of a nano-HTDMA apparatus, which can be used to measure hygroscopic growth of aerosol particles down to < 10 nm. The technique they developed is very important, and they also carried out calibration and validation experiments very comprehensively. The paper is also well-written, and I only have a few comments.*

**Response:** We are grateful to referee #1 for her/his comments and suggestions to improve our manuscript. We have implemented changes based on these comments in the revised manuscript. We repeat the specific points raised by the reviewer in italic font, followed by our response. The pages numbers and lines mentioned are with respect to the Atmospheric Measurement Techniques Discussions (AMTD) version.

*General comments:*

*(1) Compared to "sizing accuracy", "sizing offset" may better describe the actual content of Section 3.1.1. Sections 2.2.1 and Section 3.1.1: I think both sizing accuracy (difference between actual size and the size measured using a DMA) and sizing offset (i.e. measured difference between the two DMAs) are important for H-TDMA. While sizing offset has been carefully characterized (Section 3.1.1) for particles down to a few nm, not much information has been provided for the sizing accuracy for <100 nm particles. Although experiments to determine size accuracy for <100 nm particles seem to be impossible, as discussed in Section 2.2.1, could the author estimate the sizing accuracy from a theoretical view?*

**Response:** Good comment, and thanks. Yes, the reviewer is right, it is not possible to determine size accuracy for < 100 nm particles, and sub-20 nm PSL is even not available. Following the reviewer's suggestion, here we try to estimate the sizing accuracy in this size range through error propagation by using differential mobility analysis (DMA) transfer function and the uncertainties of its input parameters (Duplissy et al., 2009; Wiedensohler et al., 2012). According to Knutson and Whitby (1975), sizing of DMA transfer function mainly depends on sheath flow rates and high voltage (HV) applied to the DMA as follows:

$$z_p^* = \frac{Q_{sh} ln\frac{r_2}{r_1}}{2\pi LV} \tag{R1}$$

$$z_p^* = \frac{neC_c}{3\pi\mu d_p^*} \tag{R2}$$

$$d_p^* = \frac{2VLneC_c}{3\mu Q_{sh}ln\frac{r_2}{r_1}} \tag{R3}$$

where $z_p^*$ is the central electrical mobility, $Q_{sh}$ is the sheath flow rate, $V$ is the applied voltage, $L$ is the length of the classification region within the DMA, and $r_1$ and $r_2$ are the inner and outer radii of the DMA annulus, respectively. $n$ is the number of elementary charges of particles. $e$ is the elementary charges. $C_c$ is the slip correction. $\mu$ is the flow viscosity. $d_p^*$ is the mean particle mobility diameter.

According to Eq. (R3) above, we use the following error propagation formula (Eq. (R4)) (Taylor and Taylor, 1997) to calculate the uncertainties in sizing of nanoparticles. In our study, the flow accuracy of mass flow meter (TSI series 4000) is within ±2%. The deviation of voltage applied to the nano-DMAs (0-12500 V, 0-350 V) varies around the set value when test with voltage power supply (HCE 0-12500, HCE 0-350, Fug Electronic) shown in Table R1. Thence, the sizing accuracy is obtained using Eq. (R5) as shown in Table R1.

$$\delta z = \sqrt{\left(\frac{\partial z}{\partial x}\right)^2 (\delta x)^2 + \left(\frac{\partial z}{\partial y}\right)^2 (\delta y)^2} \tag{R4}$$

$$\frac{\delta d}{d} = \sqrt{\left(\frac{\delta V}{V}\right)^2 + \left(\frac{\delta Q_{sh}}{Q_{sh}}\right)^2} \tag{R5}$$

**Table R1 (new Table S5 in revised SI).** Uncertainties of nano-DMA voltage (V) and sheath flow rates ($Q_{sh}$), and calculated size uncertainty.

| Size (nm) | Uncertainties in V and $Q_{sh}$ | Uncertainty (Sizing accuracy) |
|---|---|---|
| 100 | 2648.2±0.02592 V, 10±0.02 L/min | 0.2000% |
| 60 | 1063.0±0.02686 V, 10±0.02 L/min | 0.2000% |
| 20 | 131.1±0.01519 V, 10±0.02L/min | 0.2003% |
| 10 | 33.7±0.02435 V, 10±0.02 L/min | 0.2127% |
| 8 | 21.6±0.03725 V, 10±0.02 L/min | 0.2641% |
| 6 | 12.2±0.06920 V, 10±0.02 L/min | 0.6014% |

**Related additions and changes included in the revised manuscript:**

**Page 13 line 299, we add:** "As discussed in Sec. 2.2.1, it is difficult to verify the sizing accuracy of sub-100 nm aerosol nanoparticles using PSL nanoparticles. Duplissy et al. (2009) and Wiedensohler et al. (2012) suggested to estimate the sizing accuracy of sub-100 nm nanoparticles through DMA transfer function. The theoretical DMA transfer function (see SI. S2. Eq. (S2-S4)) was proposed by Knutson and Whitby (1975) and they noted that sizing is crucially dependent on flow rates and high voltage (HV) applied to the DMA. In our study, the flow accuracy calibrated by the mass flow meter (TSI series 4000) is within ±2%. The variation of voltage applied to the nano-DMAs (0-12500 V, 0-350 V) around the set value were measured with voltage power supply (HCE 0-12500, HCE 0-350, Fug Electronic) and summarized in Table S5. According to the error propagation formula (see SI. S2. Eq. (S5)) (Taylor and Taylor, 1997), the calculated uncertainty in sizing of 6-100 nm nanoparticles increases as size decreases (Table S5).The estimated sizing accuracy is slightly smaller than the sizing offset of two nano-DMAs, but in principle they are still consistent with each other. This suggests that uncertainties of slip correction, DMA dimensions (inner and outer radius, length), temperature, pressure, and viscosity of air may also affect the sizing accuracy (see SI. S2. Eq. (S4), Kinney et al., 1991). Besides, Wiedensohler et al. (2012) also suggested that particle losses, the size- and material-dependent CPC counting efficiency can affect the size accuracy of DMAs."

**Related additions included in the supplementary information:**

**Line 156, we add:**

**S2. Calculation of sizing accuracy of sub-100 nanoparticles**

Knutson and Whitby (1975) proposed the following theoretical differential mobility analyzer (DMA) transfer function and showed that sizing is crucially dependent on sheath flow rates and high voltage (HV) applied to the DMA.

$$z_p^* = \frac{Q_{sh} ln\frac{r_2}{r_1}}{2\pi L V} \tag{S2}$$

$$z_p^* = \frac{neC_c}{3\pi\mu d_p^*} \tag{S3}$$

$$d_p^* = \frac{2VLneC_c}{3\mu Q_{sh} ln\frac{r_2}{r_1}} \tag{S4}$$

where $z_p^*$ is the central electrical mobility, $Q_{sh}$ is the sheath flow rate, $V$ is the applied voltage, $L$ is the length of the classification region within the DMA, and $r_1$ and $r_2$ are the inner and outer radii of the DMA annulus, respectively. $n$ is the number of elementary charges of particles. $e$ is the elementary charges. $C_c$ is the slip correction. $\mu$ is the flow viscosity. $d_p^*$ is the mean particle mobility diameter.

According to Eq. (S4) above, we use the following error propagation formula ((Taylor and Taylor, 1997) to calculate the uncertainties in sizing of nanoparticles. In our study, the flow accuracy of mass flow meter (TSI series 4000) is within ±2%. The deviation of voltage applied to the nano-DMAs (0-12500 V, 0-350 V) varies around the set value when test with voltage power supply (HCE 0-12500, HCE 0-350, Fug Electronic) shown in Table S5. Thence, the uncertainties in sizing of nanoparticles are obtained based on the following Eq. (S5) as shown in Table S5.

$$\frac{\delta d}{d} = \sqrt{\left(\frac{\delta V}{V}\right)^2 + \left(\frac{\delta Q_{sh}}{Q_{sh}}\right)^2} \qquad (S5)$$

*(2) Line 300-305: It is interesting to find that sizing offset (<0.9%) is smallest at 8 and 10 nm, smaller than that at smaller diameter (6 nm) and at larger diameter (20 nm or larger). Is there any explanation.*

**Response:** Thanks for the comment. Uncertainties in the sheath flow rates and nano-DMA voltages will increase as size decreases, which results in a larger size offset of 6-nm nanoparticles compared with other sizes. However, we observed that the peak diameter of number size distribution of the generated pure water is ~20-30 nm (Figure R1), which is more likely due to presence of impurities in the water. This interferes the accurate measurement of 20-nm nanoparticles.

[Figure]

**Figure R1.** Number concentration scanned for water nanoparticles by the nano-DMA2 at RH below 5 % at 298 K.

**Page 13 line 305, we add**: "As discussed above, uncertainties in the sheath flow rates and nano-DMA voltages will increase as size decreases, which results in a larger sizing offset of 6-nm nanoparticles compared with other sizes."

*Technical comments:*

*(1) Line 57: change "challenge" to "challenging".*
**Response:** Many thanks. We have revised in the following sentence and now they read as:
**Page 3 line 55-57:** "In addition, by knowing the hygroscopicity of newly formed nanoparticle, one can infer the involving chemical species (e.g., organic ratio) in particle formation and initial growth (Wang et al., 2010), which is otherwise difficult and highly challenging to measure directly (Wang et al., 2010; Ehn et al., 2014)."

*(2) Line 349-353: I am not sure Wikipedia is a reliable source for physical/chemical constants. I would recommend textbooks/handbooks instead.*
**Response:** Thanks for your suggestions. We have cited Atkins et al. (2006) in the following sentence:
**Page 15 line 349-353:** "It may due to the heat produced from the inner electrode of nano-DMA2, which we estimated to be ~0.08 W ($Q = mdTC_{p,}$) by considering the density and heating capacity of air, and aerosol and sheath air flow rate ($\rho$=1.2041kg/m$^3$; $C_p$=1.859kJ/kg°C) (Atkins et al., 2006)."

**Reference:**

Atkins, P., De Paula, J., and Walters, V.: Physical Chemistry, W. H. Freeman, 2006.

Kinney, P. D., Pui, D. Y. H., Mullholland, G. W. & Bryner, N. P. Use of the Electrostatic Classification Method to Size 0.1 μm SRM Particles—A Feasibility Study. Journal of Research of the National Institute of Standards and Technology, 96, 147, 1991.

Knutson, E. O. and Whitby, K. T.: Aerosol classification by electric mobility: apparatus, theory, and applications, Journal of Aerosol Science, 6, 443-451, 1975.

Taylor, J. R. and Taylor, S. L. L. J. R.: Introduction To Error Analysis: The Study of Uncertainties in Physical Measurements, University Science Books, 1997.

Wiedensohler, A., Birmili, W., Nowak, A., Sonntag, A., Weinhold, K., Merkel, M., Wehner, B., Tuch, T., Pfeifer, S., Fiebig, M., Fjäraa, A. M., Asmi, E., Sellegri, K., Depuy, R., Venzac, H., Villani, P., Laj, P., Aalto, P., Ogren, J. A., Swietlicki, E., Williams, P., Roldin, P., Quincey, P., Hüglin, C., Fierz-Schmidhauser, R., Gysel, M., Weingartner, E., Riccobono, F., Santos, S., Grüning, C., Faloon, K., Beddows, D., Harrison, R., Monahan, C., Jennings, S. G., O'Dowd, C. D., Marinoni, A., Horn, H. G., Keck, L., Jiang, J., Scheckman, J., McMurry, P. H., Deng, Z., Zhao, C. S., Moerman, M., Henzing, B., de Leeuw, G., Löschau, G., and Bastian, S.: Mobility particle size spectrometers: harmonization of technical standards and data structure to facilitate high quality long-term observations of atmospheric particle number size distributions, Atmos. Meas. Tech., 5, 657-685, 2012.

*Response to comments by anonymous referee #2:*

*This manuscript "Nano-hygroscopicity tandem differential mobility analyzer (nano-HTDMA) for investigating hygroscopic properties of sub-10 nm aerosol nanoparticles" presents a design of a HTDMA to measure the hygroscopicity particle down to ~6 nm. The performance and the methods to calibrate and validate the setup were also reported. This setup was shown to have low sizing offset (<1.4% for 100 nm particle). High accuracy for flow rates of aerosol and sheath flow (±1%) and high accuracy for voltages applied to DMA (±0.1%) were found to be crucial to achieve the low sizing offset. Also the DMA2 and humidification system were designed to be placed in housing with stable temperate (±0.1K). The RH of sheath flow was set to the same as RH of aerosol flow to prevent the pre-deliquescence. Using this setup, the authors measured the deliquescence and the efflorescence RH as well as the growth factors of ammonium sulfate and sodium sulfate. For ammonium sulfate, no significant size dependence of DRH and ERH was observed while clear size dependence was observed. Determining the hygroscopicity of nano-particles is important to understand aerosol-water interaction and provides constraints on the chemical composition of nano-particles. This nano-HTDMA has excellent performance and will be useful to measure hygroscopictiy of atmospheric nano-particles. The manuscript is well-written and fit well the scope of AMT. I recommendation its publication in AMT after addressing the following minor comments.*

**Response:** We are grateful to referee #2 for the comments and the constructive suggestions. We address in the following the comments and suggestions by referee #2 and provide improvements based on these clarify the questioned issues in the revised manuscript. The pages numbers and lines mentioned are with respect to the Atmospheric Measurement Techniques Discussions (AMTD) version.

*Minor comments:*

*(1) What is the smallest particle size that the HTDMA can measure?*

*The title "Nano-hygroscopicity tandem differential mobility analyzer (nano-HTDMA) for investigating hygroscopic properties of sub-10 nm aerosol nanoparticles" reads a little redundant for me. In addition, the manuscript discusses many experiments for particle >10 nm. I suggest optimizing the title.*

**Response:** Thanks for the comment. At the moment, the smallest size that we can measure is 6 nm. The main purpose of the instrument development is to have a device that is able to measure hygroscopic growth of sub-10 nm nanoparticles. We discussed that the results of 20 nm and 100 nm are to compare with literature studies, which are the most abundant (especially for 100 nm) and also are to demonstrate the differences between measuring hygroscopic growth of sub-10 nm nanoparticles and larger ones. Following the suggestion, we revised the tittle as "Nano-HTDMA for investigating hygroscopic properties of sub-10 nm aerosol nanoparticles".

**Related additions and changes included in the revised manuscript:**

**Page 1 line 1-2:** "Nano-HTDMA for investigating hygroscopic properties of sub-10 nm aerosol nanoparticles".

*(2) Line 428-434, and Fig. 8d, the same method, electrospray was used to generate aerosol <20 nm in this study and the study by Biskos et al. 2006. But the results (growth factors) are still different. Can the authors discuss the difference? Is it possible to generate particles of the same size, i.e. 20 nm with different methods and compare the GF?*

**Response:** Thanks for the comment.

The morphology of particles may affect their hygroscopic behavior (Mikhailov et al., 2004, 2009). Iskandar et al. (2003) and Wang et al. (2019) show that the morphology of the aerosol particles mainly depends on initial properties of droplets (e.g., chemical composition and solution concentration) and drying process. In Table 1, we compared the generation conditions with Biskos et al. 2006b for 6-10 nm ammonium sulfate nanoparticles using an electrospray. Different from generation conditions in Biskos et al. (2006b) for 6-10 nm ammonium sulfate nanoparticles, in our study, in order to minimize the multiple charged nanoparticles, three different concentrations are used so that the size (e.g., 6, 8, 10 nm) selected by the nano-DMA1 was always slight larger than peak of the number size distribution of the generated nanoparticles by the electrospray. This is to ensure that we could have as many as nanoparticles as possible to compensate the strong nanoparticle losses in the nano-HTDMA system. Besides different generation conditions shown in Table R1, the drying rate is mainly dependent on drying flow rates in the HTDMA system (Wang et al., 2019). The RH of dried ammonium sulfate aerosol nanoparticles varies due to the different aerosol/sheath flow rates employed in Biskos et al. (2006b) and this study, respectively. These differences may lead to the small difference in growth factor of ammonium sulfate nanoparticles prior to the deliquescence.

Following reviewer's suggestion, we used an electrospray and an atomizer to generate 20-nm ammonium sulfate aerosol nanoparticles, respectively. We then compared their hygroscopic growth factors prior to deliquescence. Figure R1a shows a ~ 0.1 higher growth factor of 20-nm ammonium sulfate generated by an electrospray than that using an atomizer in the RH range from 55% to 82%. Figure R1b shows the results of 20-nm sodium chloride nanoparticles using an electrospray and a vaporization-condensation method (Biskos et al., 2006a), respectively. Also, There is a slight difference in the growth factor of 20-nm sodium chloride at RH between 20% and 60% using the different generation methods.

**Table R1**. Comparison of generation of ammonium sulfate (AS) nanoparticles with diameter from 6-10 nm with Biskos et al. (2006b) using an electrospray

| Generation of 6-10 nm AS nanoparticles | AS concentration (mM) | Size of capillary | Flow rates | RH of generated AS nanoparticles |
|---|---|---|---|---|
| Biskos et al. (2006b) | 10 | 40 μm | 2 l/min dry air | 0.1% |
| This study | 1, 5, 20 | 20 μm | 1 l/min dry $N_2$ | 2% |

[Figure]

[Figure]

**Figure R1 (new Figure S12 in revised SI)**. Hygroscopic growth factors of 20-nm **(a)** ammonium sulfate from our study (AS) nanoparticles and **(b)** sodium chloride (NaCl) nanoparticles from Biskos et al. (2006a) using the different generation methods prior to deliquescence.

**Page 19 line 432, we add**: "Different from generation conditions of for 6-10 nm ammonium sulfate nanoparticles in Biskos et al. (2006b), in our study, in order to minimize the multiple charged nanoparticles, three different concentrations are used so that the size selected by the nano-DMA1 (i.e., 6, 8, 10 nm) was always slight larger than peak of the number size distribution of the generated nanoparticles by the electrospray. This also helps us to have as many as nanoparticles as possible to compensate the strong nanoparticle losses in the nano-HTDMA system. In addition, we used both electrospray and atomizer to generate 20-nm ammonium sulfate, and compared their hygroscopic growth factors prior to deliquescence. Figure S12a shows a ~ 0.1 higher growth factor of 20-nm ammonium sulfate generated by the electrospray than that using the atomizer in the RH range from 55% to 82%, which is similar to the difference in hygroscopic growth factor of 20-nm NaCl aerosol nanoparticles using the different generation methods as observed in Fig S12b in Biskos et al. (2006a). Besides different generation conditions, the morphology of dried ammonium sulfate particles may also differ slightly between our study and Biskos et al. (2006) because of different dying rates, as drying flow rates and RH of the dried ammonium sulfate in the two HTDMA systems are different too."

*(3) Fig. 5 and Fig. 7, can the author discuss why the 6 nm AS showed a slight increase with increasing RH.*

**Response:** Thanks for the comments. Yes, a slight increase in hygroscopic growth factor of 6-nm ammonium sulfate nanoparticles was observed in the RH range from 65 to 79% RH before deliquesces. This is attributed to water adsorption onto the surfaces of these nanoparticles. It seems that there is more water adsorption onto the small nanoparticles than that of large nanoparticles. Similar phenomenon has also observed by Hämeri et al. (2000, 2001), Romakkaniemi et al. (2001), Biskos et al. (2006a, b, 2007), and Giamarelou et al. (2018). The reason for such enhanced adsorption at smaller sizes is still to be investigated.

**Page 19 line 428, we added:** "For example, a slight increase in hygroscopic growth factor of 6-nm ammonium sulfate nanoparticles is observed in the RH range from 65 to 79% RH before deliquescence. This is attributed to water adsorption onto the surfaces of these nanoparticles. It seems that smaller nanoparticles have a stronger tendency of adsorbing water when approaching the DRH than the larger ones. Similar phenomenon has also observed by Hämeri et al. (2000, 2001), Romakkaniemi et al. (2001), Biskos et al. (2006a, b, 2007), and Giamarelou et al. (2018). The reason for such enhanced adsorption at smaller sizes is still to be investigated."

*(4) Fig.7, why the DRH for 20 nm AS is different from others (the dashed line)? Also the coloring of efflorescence and deliquescence in this panel contradicts the caption.*

**Response:** Thanks for the comment. The DRH of 20-nm ammonium sulfate is slightly different from that at other sizes. Also, the similar phenomenon was observed for 20-nm ammonium sulfate nanoparticles from Biskos et al. (2006b) shown in Fig. R2, which shows in agreement with our study. To my knowledge, we observed that the peak diameter of number size distribution of pure water is ~20-30 nm (Figure S2a), which is more likely due to presence of impurities in the water. This interferes the accurate measurement of 20-nm nanoparticles.

[Figure]

**Figure R2**. Comparison of the hygroscopic behavior of 20-nm ammonium sulfate (AS) with Biskos et al. (2006b).

**Page 45 line 917, we revised the color of 20-nm ammonium sulfate in both deliquescence and efflorescence measurement modes and made the color consistent with citation:**

[Figure]

**Figure 7.** Mobility-diameter hygroscopic growth factors ($g_f$) of ammonium sulfate (AS) aerosol nanoparticles with dry mobility diameter from 6 to 100 nm in the deliquescence mode (red square and error bar) and the efflorescence mode (royal square and error bar). Deliquescence, and efflorescence relative humidity (DRH&ERH, black dashed line) of ammonium sulfate (AS) nanoparticles with dry mobility diameter from 6 to 100 nm.

*(5) Line 375-376, "double-mode phenomenon was not observed 375 for 8 and 6 nm ammonium sulfate nanoparticles". Is this because of the slower mass transfer of water vapor for larger particles?*

**Response:** Many thanks. No, this is not because of the slower mass transfer of water vapor for larger sulfate nanoparticles. Double-mode phenomenon was observed for 99-nm sodium chloride in Mikhailov et al. (2004) and for 10-nm ammonium sulfate and sodium chloride in Biskos et al. (2006b, 2007) in the deliquescence measurement mode, respectively. They attributed this to the co-existence of solid and liquid phase of aerosol nanoparticles due to the slight inhomogeneity of RH within nano-DMA2. Bezantakos et al. (2016) have shown the difference of RH for sheath flow and aerosol flow upstream of DMA2 and temperature gradient within DMA2 can result in RH non-uniformities within DMA2. In our study, we also observed this double-mode phenomenon for ammonium sulfate nanoparticles with diameters (e.g., 100, 60, 20, 10 nm) but not for 8 and 6 nm ammonium sulfate nanoparticles. Because this phenomenon is an essentially stochastic process.

*(6) Line 472-474, why does DRH/ERH of sodium sulfate show a clear size dependence while ammonium sulfate does not?*

**Response:** Many thanks.

Different from ammonium sulfate, of which DRH and ERH shows no significant size dependence, there is a strong size-dependence of DRH and ERH of sodium sulfate according to our observations down to 6 nm. The different size dependence of DRH and ERH between sodium chloride and ammonium sulfate have been theoretically studied and explained by Cheng et al. (2015). The main reason is the different concentration dependence of solute activities and the different solute-liquid surface tension, e.g., the same change in solute molality leads to a larger change in the solute activity of sodium chloride than that of ammonium sulfate shown in Fig. R3. The phase transition concentration (deliquescence and crystallization concentration) of ammonium sulfate is thus more sensitive to the size change compared to that of sodium chloride, leading to the almost unchanged DRH and ERH of ammonium sulfate nanoparticles (Cheng et al., 2015). For the size dependence of phase transition of sodium sulfate, there is a clear size effect on DRH and ERH similar to that of sodium chloride but different from that of ammonium sulfate in the size range from 6 to 20 nm, suggesting that non-ideality of solution property is close to that of sodium chloride but weaker than that of ammonium sulfate.

[Figure]

**Figure R3**. Saturation ratio of solute activity ($a_s^*/a_{s,bulk}^*$) as a function of molality $b$ for ammonium sulfate (AS) and sodium chloride (NaCl). Reprinted with permission by Cheng et al. (2015).

**Page 20 line 466-477, we revised:** "The strong size-effect on the DRH and ERH of sodium chloride and on hygroscopic growth factors of ammonium sulfate have been observed by Biskos et al. (2006a, b, 2007) and theoretically studied and explained by Cheng et al. (2015). Owning to the strong non-ideality of aqueous ammonium sulfate solution, the phase transition concentration (deliquescence and crystallization concentration) of ammonium sulfate is much more sensitivity to the size changes from 60 nm to 6 nm than that of sodium chloride, leading to the almost unchanged DRH and ERH of ammonium sulfate nanoparticles (Cheng et al., 2015). Compared the three compounds, the size-dependent hygroscopicity of sodium sulfate nanoparticles from 20 nm to 6 nm is similar to that of sodium chloride, but different to that of ammonium sulfate, where no significant change in DRH and ERH was observed. However, in this size range, the increase of the ERH and the decrease of growth factor upon decreasing size seems to be stronger for sodium sulfate than sodium chloride, although no significant change in DRH was observed from micrometer size particles down to 20 nm." **as**

"Different from ammonium sulfate, of which DRH and ERH shows no significant size dependence, there is a strong size-dependence of DRH and ERH of sodium sulfate according to our observations down to 6 nm. The different size dependence of DRH and ERH between sodium chloride and ammonium sulfate have been theoretically studied and explained by Cheng et al. (2015). The main reason is the different concentration dependence of solute activities and the different solute-liquid surface tension, e.g., the same change in solute molality leads to a larger change in the solute activity of sodium chloride than that of ammonium sulfate. The phase transition concentration (deliquescence and crystallization concentration) of ammonium sulfate is thus more sensitive to the size change compared to that of sodium chloride, leading to the almost unchanged DRH and ERH of ammonium sulfate nanoparticles (Cheng et al., 2015). For the size dependence of phase transition of sodium sulfate, a strong size effect on DRH and ERH is similar to that of sodium chloride but different from that of ammonium sulfate in the size range from 6 to 20 nm, suggesting that non-ideality of solution property is close to that of sodium chloride but weaker than that of ammonium sulfate."

***Technical comments:***

*(1). Line 347, "excuses air" or "excess air"?*

**Response:** Many thanks. We have carefully checked and revised the whole of manuscript and supplement information, including grammar, wording, and sentence structure.

**Page 15 line 347-349:** "we monitored that the sheath flow temperature at the inlet of nano-DMA2 is slightly lower (less than ~0.2 K) than that at the outlet, i.e., the RHs at the inlet of nano-DMA2 is slightly higher (~ 1%) than the RH of the excess air at the outlet."

*(2). Line 427, "continues" should be "continuous".*

**Response:** Many thanks. We have revised in the following sentence and now they read as:

**Page 18 line 426-428:** "There seems to be continuous water adsorption and the adsorbed water layers (Romakkaniemi et al., 2001) become significantly thicker when RH closer to the DRH (i.e, RH > 70%)."

*(3). Line 470, "sensitivity" should be "sensitive".*

**Response:** Many thanks. We have revised in the following sentence and now they read as:

**Page 20 line 468-472:** "The phase transition concentration (deliquescence and crystallization concentration) of ammonium sulfate is thus more sensitive to the size change compared to that of sodium chloride, leading to the almost unchanged DRH and ERH of ammonium sulfate nanoparticles (Cheng et al., 2015)."

*(4). Fig. 5 and 6, I suggest explaining the red and blue lines in the captions, although they were explained in the main text.*

**Response:** Many thanks. We add explanations of the red and blue lines in the all captions in the manuscript and supplement information, respectively.

**Page 44 line 907-909: "Figure 5.** Deliquescence-mode measurements of ammonium sulfate (AS) aerosol nanoparticles with dry mobility diameter from 20-6nm. The measured (black square) and fitted (solid lines) normalized size distribution are shown for increasing RH. The red and blue lines represent the aerosol nanoparticles in the solid and liquid state, respectively. The RH history in each measurement is 5% → X%, where X is the RH value given in each panel."

**Page 45 line 912-914: "Figure 6.** Efflorescence-mode measurements of ammonium sulfate (AS) aerosol nanoparticles with dry mobility diameter from 20-6nm. The measured (black circle) and fitted (solid lines) normalized size distribution are shown for increasing RH. The red and blue lines represent the aerosol nanoparticles in the solid and liquid state, respectively. The RH history in each measurement is 5%→97%→X%, where X is the RH value given in each panel."

**Related additions and changes included in the revised supplement information:**

**Line 44-47: "Figure S4.** Deliquescence-mode **(a)** and efflorescence-mode **(b)** of 100-nm ammonium sulfate (AS) aerosol nanoparticles. The measured (black square) and fitted (solid lines) normalized size distribution are shown for increasing RH (5%→X%, where X is the RH value given in each panel) and decreasing RH (5%→97%→X%, where X is the RH value given in each panel), respectively. The red and blue lines represent the aerosol nanoparticles in the solid and liquid state, respectively."

**Line 52-55: "Figure S5.** Deliquescence-mode **(a)** and efflorescence-mode **(b)** of 60-nm ammonium sulfate (AS) aerosol nanoparticles. The measured (black square) and fitted (solid lines) normalized size distribution are shown for increasing RH (5%→X%, where X is the RH value given in each panel) and decreasing RH (5%→97%→X%, where X is the RH value given in each panel), respectively. The red and blue lines represent the aerosol nanoparticles in the solid and liquid state, respectively."

**Line 59-62: "Figure S6.** Deliquescence-mode **(a)** and efflorescence-mode **(b)** of 8-nm ammonium sulfate (AS) aerosol nanoparticles. The measured (black square) and fitted (solid lines, single-mode log-normal fit) normalized size distribution are shown for increasing RH (5%→X%, where X is the RH value given in each panel) and decreasing RH (5%→97%→X%, where X is the RH value given in each panel), respectively. The red and blue lines represent the aerosol nanoparticles in the solid and liquid state, respectively."

**Line 95-100:** "**Figure S9.** Deliquescence-mode **(a)** and efflorescence-mode **(b)** of 20-nm sodium sulfate aerosol nanoparticles. The measured (black square) and fitted (solid lines) normalized size distribution are shown for increasing RH (5%→X%, where X is the RH value given in each panel) and decreasing RH (5%→97%→X%, where X is the RH value given in each panel), respectively. Red/blue solid line is fitted by a single-mode log-normal fit. Red, blue, and black lines are fitted by a double-mode log-normal fit. The red and blue lines represent the aerosol nanoparticles in the solid and liquid state, respectively. The voltage applied to the nano-DMAs (0-12500 V) is kept within ±1% around the set value shown in the voltage meter."

**Line 104-109:** "**Figure S10.** Deliquescence-mode **(a)** and efflorescence-mode **(b)** of 6-nm sodium sulfate aerosol nanoparticles. The measured (black square) and fitted (solid lines) normalized size distribution are shown for increasing RH (5%→X%, where X is the RH value given in each panel) and decreasing RH (5%→97%→X%, where X is the RH value given in each panel), respectively. Red/blue solid line is fitted by a single-mode log-normal fit. Red, blue, and black lines are fitted by a double-mode log-normal fit. 
[revised manuscript text omitted]

**Table S1:** Deliquescence and efflorescence relative humidity of ammonium sulfate below 100 nm reported by difference studies in temperature ranging from 290-300K

| Deliquescence relative humidity (DRH) | Efflorescence relative humidity (ERH) | Technique (initial particle size) | Reference |
|---|---|---|---|
| 80-86%* (8 nm) 80-85%* (10 nm) 80-90%* (15 nm) 78-80%* (30 nm) 76-79%* (50 nm) | | HTDMA (8.10,15,30,50 nm) | Hämeri et al. (2000) (cf. Figure 2a, 2b, 2c, 2d, and 2e) |
| 76-80%* | 65%* | HTDMA (100 nm) | Gysel et al. (2002) (cf. Figure 2) |
| 82% (6 nm) 81% (8 nm) 80% (10 nm) 82% (20 nm) 80% (40 nm) 80% (60 nm) | 34% (6 nm) 33% (8 nm) 35% (10 nm) 35% (20 nm) 36% (40 nm) 33% (6 nm) | HTDMA (6,8,10,20,40,60 nm) | Biskos et al. (2006b) |
| - | 27-31%* (43.7 nm) 21-30.7%* (47 nm) | HTDMA (43.7,47 nm) | Gao et al. (2006) (cf. Figure 5) |
| 78-81%* | - | HTDMA (100 nm) | Duplissy et al. (2009) (cf. Figure 4) |
| 77-78%* | - | HTDMA | Duplissy et al. (2009) |

| | | | |
|---|---|---|---|
| | | (100 nm) | (cf. Figure 4) |
| 78-80%* | 29-34%* | HTDMA | Mikhailov et al. (2009) (cf. Fig4) |
| | | (100 nm) | |
| 77-78% | - | HTDMA | Wu et al. (2011) |
| | | (100 nm) | |

-: Not reported

*: Data retrieved from figures in the references

80-86%: Non-prompt deliquescence of 8-nm ammonium sulfate from 80% to 86% RH

27-31%: Non-prompt efflorescence of 43.7-nm ammonium sulfate from 31% to 27% RH

82%: Prompt deliquescence of 6-nm ammonium sulfate at 82% RH

**Table S2.** Residence time (s) for the water equilibrium for particles with diameter ranging from 6 to 100 nm particles at RH=90% at 25°C

| χ | 1 | 0.1 | 0.01 | 0.001 |
|---|---|-----|------|-------|
| 100nm | $6.26 \times 10^{-6}$ | $3.55 \times 10^{-5}$ | $3.12 \times 10^{-4}$ | 0.0031 |
| 60nm | $6.04 \times 10^{-6}$ | $3.34 \times 10^{-5}$ | $3.07 \times 10^{-4}$ | 0.0030 |
| 20nm | $6.03 \times 10^{-7}$ | $5.17 \times 10^{-6}$ | $5.08 \times 10^{-5}$ | $5.07 \times 10^{-4}$ |
| 10nm | $1.88 \times 10^{-7}$ | $1.74 \times 10^{-6}$ | $1.73 \times 10^{-5}$ | $1.72 \times 10^{-4}$ |
| 8nm | $3.10 \times 10^{-8}$ | $1.93 \times 10^{-7}$ | $1.82 \times 10^{-6}$ | $1.81 \times 10^{-5}$ |
| 6nm | $1.48 \times 10^{-8}$ | $1.08 \times 10^{-7}$ | $1.04 \times 10^{-6}$ | $1.03 \times 10^{-5}$ |

**Table S3.** Average sizing offset between nano-DMAs in the nano-HTDMA system at RH below 10%

| | Average sizing offset (nm)[a] | Size agreement between nano-DMA1 and nano-DMA2[b] |
|---|---|---|
| 100-nm $(NH_4)_2SO_4$ | 0.619318 | 0.619318% |
| 60-nm $(NH_4)_2SO_4$ | 0.298691 | 0.4978% |
| 20-nm $(NH_4)_2SO_4$ | 0.278311 | 1.3916% |
| 10-nm $(NH_4)_2SO_4$ | 0.089647 | 0.8965% |
| 8-nm $(NH_4)_2SO_4$ | -0.01598 | -0.19975% |
| 6-nm $(NH_4)_2SO_4$ | 0.083965 | 1.3994 % |

[a] Calculation from $(\bar{D}_{measured\ by\ nano-DMA2} - D_{selected\ by\ nano-DMA1})$

[b] Calculation from $[(\bar{D}_{measured\ by\ nano-DMA2} - D_{selected\ by\ nano-DMA1}) / D_{selected\ by\ nano-DMA1}] \times 100\%$

Formatted Table

**Table S4.** The values of $D_m$, $g_f$, and $D_m$ (< 5% RH) of 10-nm ammonium sulfate of Biskos et al.
(2006b) system in the different RHs.

| Relative humidity | $D_m$ | $g_f$ | $D_m$ (<5 % RH) |
|---|---|---|---|
| 25% | 10.3982439 | 0.992914120 | 10.47245043 |
| 76% | 10.38867117 | 1.017488426 | 10.21011237 |
| 78% | 10.54314064 | 1.027692308 | 10.25904404 |
| 80% | 13.31036607 | 1.293796610 | 10.28783502 |
| 44% | 11.56059002 | 1.120463542 | 10.31768513 |
| 35% | 11.24527292 | 1.084064417 | 10.37325157 |
| 34% | 10.59107394 | 1.007786565 | 10.50924304 |
| 32% | 10.24542551 | 1.003831854 | 10.20631639 |
| 31% | 10.20845456 | 1.001920937 | 10.18888236 |
| 30% | 10.38101934 | 1.001441750 | 10.36607405 |
| 29% | 10.27755951 | 1.003183756 | 10.2779752 |
| 24% | 10.26077112 | 0.997295121 | 10.28860053 |

**Table S5.** Uncertainties of nano-DMA voltage (V) and sheath flow rates ($Q_{sh}$), and calculated size
uncertainty.

| Size (nm) | Uncertainties in V and $Q_{sh}$ | Uncertainty (Sizing accuracy) |
|---|---|---|
| 100 | 2648.2±0.02592* V, 10±0.02* L/min | 0.2000% |
| 60 | 1063.0±0.02686 V, 10±0.02 L/min | 0.2000% |
| 20 | 131.1±0.01519 V, 10±0.02L/min | 0.2003% |
| 10 | 33.7±0.02435 V, 10±0.02 L/min | 0.2127% |
| 8 | 21.6±0.03725 V, 10±0.02 L/min | 0.2641% |
| 6 | 12.2±0.06920 V, 10±0.02 L/min | 0.6014% |

[Figure]

**Figure S1.** Methods for measuring hygroscopicity of atmospheric aerosol particles in different size ($D_p$).

[Figure]

**Figure S2. (a)** Number concentration scanned for water nanoparticles by the nano-DMA2 at RH below 5 % at 298 K.

**(b)** Normalized number size distribution scanned for 22-nm PSL nanoparticles by nano-DMA2 after calibration.

[Figure]

**Figure S3.** Number size distribution of ammonium sulfate (AS) nanoparticles (black solid square) generated by the electrospray. **(a)** 20mM, **(b)** 5mM, and **(c)** 1mM AS solution. The dotted line marks peak diameter from the Gaussian fits for the scan (red curve). The black solid lines mark the diameters of the monodispersed nanoparticles selected by the nano-DMA1.

[Figure]

**Figure S4.** Deliquescence-mode **(a)** and efflorescence-mode **(b)** of 100-nm ammonium sulfate (AS) aerosol nanoparticles. The measured (black square) and fitted (solid lines) normalized size distribution are shown for increasing RH (5%→X%, where X is the RH value given in each panel) and decreasing RH (5%→97%→X%, where X is the RH value given in each panel), respectively. The red and blue lines represent the aerosol nanoparticles in the solid and liquid state, respectively.

[Figure]

**Figure S5.** Deliquescence-mode **(a)** and efflorescence-mode **(b)** of 60-nm ammonium sulfate (AS) aerosol nanoparticles. The measured (black square) and fitted (solid lines) normalized size distribution are shown for increasing RH (5%→X%, where X is the RH value given in each panel) and decreasing RH (5%→97%→X%, where X is the RH value given in each panel), respectively. The red and blue lines represent the aerosol nanoparticles in the solid and liquid state, respectively.

[Figure]

**Figure S6.** Deliquescence-mode **(a)** and efflorescence-mode **(b)** of 8-nm ammonium sulfate (AS) aerosol nanoparticles.

The measured (black square) and fitted (solid lines, single-mode log-normal fit) normalized size distribution are shown for increasing RH (5%→X%, where X is the RH value given in each panel) and decreasing RH (5%→97%→X%, where X is the RH value given in each panel), respectively. The red and blue lines represent the aerosol nanoparticles in the solid and liquid state, respectively.

[Figure]

**Figure S7.** Mobility-diameter hygroscopic growth factors ($g_f$, black squares), deliquescence and efflorescence relative humidity (DRH&ERH, black dashed lines) of ammonium sulfate (AS)  nanoparticles with dry diameter from 6 to 100

nm, respectively. Red squares and dashed lines show the respective results from Biskos et al. (2006b).

[Figure]

[Figure]

**Figure S8. (a)** Comparison of mobility-diameter hygroscopic growth factors ($g_l$) of 100-nm (black square) with 6-nm (red square) ammonium sulfate (AS) nanoparticles. **(b)** Dependence of deliquescence and efflorescence relative humidity (DRH&ERH) of ammonium sulfate (AS) on dry volume equivalent diameter ($D_{ve}$). The measured DRH and ERH of ammonium sulfate within RH uncertainty (black line + black square) compared with data from Biskos et al. (2006b) (red square) in the volume equivalent diameter with shape factor ($\chi$=1.02) range from 5 to 100 nm.

[Figure]

**Figure S9.** Deliquescence-mode **(a)** and efflorescence-mode **(b)** of 20-nm sodium sulfate aerosol nanoparticles. The measured (black square) and fitted (solid lines) normalized size distribution are shown for increasing RH (5%→X%, where X is the RH value given in each panel) and decreasing RH (5%→97%→X%, where X is the RH value given in each panel), respectively. Red/blue solid line is fitted by a single-mode log-normal fit. Red, blue, and black lines are fitted by a double-mode log-normal fit. The red and blue lines represent the aerosol nanoparticles in the solid and liquid state, respectively. The voltage applied to the nano-DMAs (0-12500 V) is kept within ±1% around the set value shown in the voltage meter.

[Figure]

**Figure S10.** Deliquescence-mode **(a)** and efflorescence-mode **(b)** of 6-nm sodium sulfate aerosol nanoparticles. The measured (black square) and fitted (solid lines) normalized size distribution are shown for increasing RH (5%→X%, where X is the RH value given in each panel) and decreasing RH (5%→97%→X%, where X is the RH value given in each panel), respectively. Red/blue solid line is fitted by a single-mode log-normal fit. Red, blue, and black lines are fitted by a double-mode log-normal fit. The red and blue lines represent the aerosol nanoparticles in the solid and liquid state, respectively. The voltage applied to the nano-DMAs (0-350 V) is kept within ±1% around the set value shown in the voltage meter.

[Figure]

**Figure S11. (a)** Comparison of mobility-diameter hygroscopic growth factors ($g_l$) of 20-nm **(a)** and 60-nm **(b)**
ammonium sulfate (AS) nanoparticles with Biskos et al. (2006b) and Hu et al. (2010). (black squares: in this study;
red square: Biskos et al. (2006b); blue square: Hu et al. (2010)). **(c)** Comparison of mobility-diameter hygroscopic
growth factors of 20-nm $Na_2SO_4$ nanoparticles with Hu et al. (2010). (black squares: in this study; red square: Hu et
al. (2010)). **(d)** Mobility-diameter hygroscopic growth factors of $Na_2SO_4$ nanoparticles with diameter from 6 nm to
14~16 um at 84% RH (black solid squares: in this study; black open square: Hu et al. (2010); black open cycle: Tang
et al. (2007)). A fitting equation ($g_f = \frac{1.804}{1+(0.5267*D)^{-0.8194}}$) based on this study at 6-nm, 20-nm $Na_2SO_4$, and 14~16 um
data from Tang et al. (2007).

[Figure]

[Figure]

**Figure S12**. Hygroscopic growth factors of 20-nm **(a)** ammonium sulfate (AS) nanoparticles from our study and **(b)** sodium chloride (NaCl) nanoparticles from Biskos et al. (2006a) using the different generation methods prior to deliquescence of ammonium sulfate.

**S1. Calculation of sizing offset of 10-nm AS**

The mobility growth factor ($g_f$) is given by:

$$g_f = \frac{D_m(RH)}{D_m(<10 \% RH)} \tag{S1}$$

$g_f$ was from the data of Biskos et al. (2006b) in the different RHs (see the SI. Fig.5). $D_m$ was
retrieved the data of Biskos et al. (2006b) in the different RHs (see the SI. Fig.2) as follows:

[Figure]

**Figure S13.** Measured (black square) and fitted (red solid line) normalized number size distributions are show for
ammonium sulfate aerosol particles at 25% RH. The black square symbols show the data of Biskos et al. (2006b) (see
the S1. Fig. 2).

Therefore, the initial dry mobility diameter ($D_m$ (< 5% RH)) was obtained using Eq. (S1) based on values of $g_f$ and $D_m$ in the different RHs (see SI. Table S4). We further calculated the average sizing offset of 10-nm ammonium sulfate of Biskos et al. (2006b) system based on the values of $D_m$ (<

5% RH). The average sizing offset of 10-nm was ~3.1%.

**S2. Calculation of sizing accuracy of sub-100 nanoparticles**

Knutson and Whitby (1975) proposed the following theoretical differential mobility analyzer (DMA) transfer function and showed that sizing is crucially depend on sheath flow rates and high voltage (HV) applied to the DMA.

$$z_p^* = \frac{Q_{sh}ln\frac{r_2}{r_1}}{2\pi LV} \tag{S2}$$

$$z_p^* = \frac{neC_c}{3\pi\mu d_p^*} \tag{S3}$$

$$d_p^* = \frac{2VLneC_c}{3\mu Q_{sh}ln\frac{r_2}{r_1}} \tag{S4}$$

where $z_p^*$ is the central electrical mobility, $Q_{sh}$ is the sheath flow rate, $V$ is the applied voltage, $L$ is the length of the classification region within the DMA, and $r_1$ and $r_2$ are the inner and outer radii of the DMA annulus, respectively. $n$ is the number of elementary charges of particles. $e$ is the elementary charges. $C_c$ is the slip correction. $\mu$ is the flow viscosity. $d_p^*$ is the mean particle mobility diameter.

According to Eq. (S4) above, we use the following error propagation formula ((Taylor and Taylor, 1997) to calculate the uncertainties in sizing of nanoparticles. In our study, the flow accuracy of mass flow meter (TSI series 4000) is within ±2%. The deviation of voltage applied to the nano-DMAs (0-12500 V, 0-350 V) varies around the set value when test with voltage power supply (HCE 0-12500, HCE 0-350, Fug Electronic) shown in Table S5. Thence, the uncertainties in sizing of nanoparticles are obtained based on the following Eq. (S5) as shown in Table S5.

$$\frac{\delta d}{d} = \sqrt{\left(\frac{\delta V}{V}\right)^2 + \left(\frac{\delta Q_{sh}}{Q_{sh}}\right)^2} \tag{S5}$$

---

## Author Response (AR2)

*Response to comments by editor:*

*Thank you for making changes in response to the referee comments, and for an interesting and well-written manuscript. I have a conceptual question and then several technical corrections:*

**Response:** We are grateful to editor for her/his comments and suggestions to improve our manuscript. We have implemented changes based on these comments in the revised manuscript. We repeat the specific points raised by the editor in italic font, followed by our response. The pages numbers and lines mentioned are with respect to the third version that uploaded on 16.08.2020.

*Technical comments:*

*1) Line 367: I don't understand the heat source from the center electrode. Are you assuming it is at different temperature than the air, and if so, why? Please explain all the variables in the equation.*

**Response:** Yes, we assume that the inner electrode is at different temperature than the air. We observed that temperature of excess flow is ~0.2 °C higher than that of sheath flow at the inlet of nano-DMA2, while temperature of sheath flow is equal to that of aerosol flow at the inlet of nano-DMA2 during the measurements. Thence, a small temperature difference within nano-DMA2 is more likely due to the heat transfer between the inner electrode and air which flows around it by convection/conduction (Bezantakos et al., 2016). The plausible reason could be that when charged nanoparticles (similar to the electric current) hit the inner electrode, the inner electrode has some resistive heating from the electric current that flows. Such temperature difference/gradient within DMA was observed by previous studies (Biskos et al., 2007; Villani et al., 2008; Dupplissy et al., 2009; Bezantakos et al., 2016; Giamarelou et al., 2018). For example, a ±0.5 °C temperature difference within DMA was observed by Giamarelou et al. (2018) during the measurements. Except for the possibly slightly higher temperature of the inner electrode than the surrounding air, temperature gradient in DMA2 may also be caused by environmental disturbance or temperature difference between other parts of DMA and between sheath flow and aerosol flow.

Thus, we used the following heat capacity equation to calculate the change in heat of a nano-DMA2 system ($Q$) at a constant pressure.

$$Q = mdT C_{p,k} \tag{1}$$

Here, $m$ is mass of air flow (sheath and aerosol air flow), $dT$ is infinitesimal temperature change within a nano-DMA2 system, and $C_{p,k}$ is specific heat capacity of air. The heat produced from the inner electrode of nano-DMA2 is ~0.08 W.

**Related changes included in the revised manuscript:**

**Page 16 line 363-369:** we revised "Unlike previously reported by Bezantakos et al. (2016) that the RH at the outlet was higher than that the inlet of the sheath air, we monitored that the sheath flow temperature at the inlet of nano-DMA2 is slightly lower (less than ~0.2 K) than that at the outlet, i.e., the $RH_s$ at the inlet of nano-DMA2 is slightly higher (~ 1%) than the RH of the excess air at the outlet. It may due to the heat produced from the inner electrode of nano-DMA2, which we estimated to be ~0.08 W ($Q = mdTC_{p,k}$) by considering the density and heating capacity of air, and aerosol and sheath air flow rate (ρ=1.2041kg/m$^3$; $C_p$= 1.859kJ/kg°C) (Atkins et al., 2006)." **as** "Unlike previously reported by Bezantakos et al. (2016) that the RH at the outlet was higher than that the inlet of the sheath air, we monitored that the sheath flow temperature at the inlet of nano-DMA2 is slightly lower (less than ~0.2 K) than that at the outlet (i.e., the $RH_s$ at the inlet of nano-DMA2 is slightly higher (~ 1%) than the RH of the excess air at the outlet), while temperature of sheath flow is equal to that of aerosol flow at the inlet of nano-DMA2 during the measurements. A small temperature difference within nano-DMA2 is more likely due to the heat transfer between the inner electrode and air which flows around it by convection/conduction (Bezantakos et al., 2016). The plausible reason could be that when charged nanoparticles (similar to the electric current) hit the inner electrode, the inner electrode has some resistive heating from the electric current that flows. Such temperature difference/gradient within DMA was observed by previous studies (Biskos et al., 2007; Villani et al., 2008; Dupplissy et al., 2009; Bezantakos et al., 2016; Giamarelou et al., 2018). For example, a ±0.5 °C temperature difference within DMA was observed by Giamarelou et al. (2018) during the measurements. Except for the possibly slightly higher temperature of the inner electrode than the surrounding air, temperature gradient in DMA2 may also be caused by environmental disturbance or temperature difference between other parts of DMA and between sheath flow and aerosol flow. In this study, we calculate the change in heat ($Q$) of a nano-DMA2 system at a constant pressure, which estimates to be ~0.08 W ($Q = mdTC_{p,k}$) by considering the density and heating capacity of air, and aerosol and sheath air flow rate (ρ=1.2041kg/m$^3$; $C_p$= 1.859kJ/kg°C) (Atkins et al., 2006)."

*2) Fig. 3 caption: The sentence regarding the black solid lines is repeated at the end of the caption.*

**Response:** Many thanks. We have deleted this sentence and now they read as:

**Page 43 line 918-924:** "**Figure 3**. Sizing accuracy and sizing offset of nano-DMAs after calibration. **(a)** Normalized number size distribution scanned by the nano-DMA2 for 100-nm PSL nanoparticles (black solid square). Normalized number size distributions scanned by the nano-DMA2 for 100-nm PSL nanoparticles **(b)**, 60-nm **(c)**, and 10-nm **(d)** ammonium sulfate (AS) selected by the nano-DMA1 at RH below 5% at 298 K (black solid square). The dotted lines mark the diameters of the monodispersed nanoparticles selected by the nano-DMA1, i.e., 100 nm in **(b)**, 60 nm in **(c)** and 10 nm in **(d)**. The black solid lines mark the peak diameters from the Gaussian fits (red curve)."

*3) Tables S3, S4, and S5: These tables have far more significant figures than are plausible. Please use an appropriate number of sig figs (maybe 3?).*

**Response:** Many thanks. We have revised in the following tables accordingly.

**Related changes included in the revised supplement:**

**Table S3.** Average sizing offset between nano-DMAs in the nano-HTDMA system at RH below 10%

| | Average sizing offset (nm)[a] | Size agreement between nano-DMA1 and nano-DMA2[b] |
|---|---|---|
| 100-nm $(NH_4)_2SO_4$ | 0.619 | 0.619% |
| 60-nm $(NH_4)_2SO_4$ | 0.299 | 0.498% |
| 20-nm $(NH_4)_2SO_4$_ | 0.278 | 1.39% |
| 10-nm $(NH_4)_2SO_4$ | 0.0896 | 0.897% |
| 8-nm $(NH_4)_2SO_4$ | -0.0160 | -0.200% |
| 6-nm $(NH_4)_2SO_4$_ | 0.0840 | 1.40% |

[a] Calculation from $(\bar{D}_{measured\ by\ nano-DMA2} - D_{selected\ by\ nano-DMA1})$

[b] Calculation from $[(\bar{D}_{measured\ by\ nano-DMA2} - D_{selected\ by\ nano-DMA1})/ D_{selected\ by\ nano-DMA1}] \times 100\%$

**Table S4.** The values of $D_m$, $g_f$, and $D_m$ (< 5% RH) of 10-nm ammonium sulfate of Biskos et al. (2006b) system in the different RHs.

| Relative humidity | $D_m$ | $g_f$ | $D_m$ (<5 % RH) |
|---|---|---|---|
| 25% | 10.4 | 0.991 | 10.5 |
| 76% | 10.4 | 1.02 | 10.2 |
| 78% | 10.5 | 1.03 | 10.3 |
| 80% | 13.3 | 1.29 | 10.3 |
| 44% | 11.6 | 1.12 | 10.3 |
| 35% | 11.2 | 1.08 | 10.4 |
| 34% | 10.6 | 1.01 | 10.5 |
| 32% | 10.2 | 1.00 | 10.2 |
| 31% | 10.2 | 1.00 | 10.2 |
| 30% | 10.4 | 1.00 | 10.4 |
| 29% | 10.3 | 1.00 | 10.3 |
| 24% | 10.3 | 0.997 | 10.3 |

**Table S5.** Uncertainties of nano-DMA voltage (V) and sheath flow rates ($Q_{sh}$), and calculated size uncertainty.

| Size (nm) | Uncertainties in V and $Q_{sh}$ | Uncertainty (Sizing accuracy) |
|---|---|---|
| 100 | $2.65 \times 10^3 \pm 0.0259$ V, $10 \pm 0.0200$ L/min | 0.200% |
| 60 | $1.06 \times 10^3 \pm 0.0269$ V, $10 \pm 0.0200$ L/min | 0.200% |
| 20 | $1.31 \times 10^2 \pm 0.0152$ V, $10 \pm 0.0200$ L/min | 0.200% |
| 10 | $3.37 \times 10^1 \pm 0.0244$ V, $10 \pm 0.0200$ L/min | 0.213% |
| 8 | $2.16 \times 10^1 \pm 0.0373$ V, $10 \pm 0.0200$ L/min | 0.264% |
| 6 | $1.22 \times 10^1 \pm 0.0692$ V, $10 \pm 0.0200$ L/min | 0.601% |

*4) References: The formatting is very uneven, a consequence of using EndNote-type software. Please consistently abbreviate journal names, do not capitalize article titles, and please follow the other Copernicus formatting requirements.*

**Response:** Many thanks. We have checked and revised all reference formats according to the Copernicus formatting requirements, and revised references are marked with red in the manuscript.

*5) Line 36: Change "interests" to "interest"*

**Response:** Many thanks. We have revised in the following sentence and now they read as:

**Page 2 line 36-37:** "The climatic effects of aerosol nanoparticles have attracted increasing interest in recent years (Wang et al., 2016; Andreae et al., 2018; Fan et al., 2018)."

*6) Line 68: Change "focus" to "focused"*

**Response:** Many thanks. We have revised in the following sentence and now they read as:

**Page 3 line 68-71:** "Using these techniques, most of the early lab studies focused on the hygroscopic behavior of particles in accumulation modes and super-micron size range, including deliquescence, efflorescence of pure components and the effect of organics on the change or suppression of deliquescence and efflorescence of these inorganic components in mixtures."

*7) Line 72: Change "attempting" to "that have attempted"*

**Response:** Many thanks. We have revised in the following sentence and now they read as:

**Page 3 line 72-75:** "For nanoparticles with diameters down to sub-10 nm, there are, however, only very few studies that have attempted to investigate their interactions with water molecules, which mainly utilized the setup with humidified tandem DMAs (Hämeri et al., 2000, 2001; Sakurai et al., 2005; Biskos et al., 2006a, b, 2007; Giamarelou et al., 2018)."

*8) Line 87: Change "pre-deliquesced" to "pre-deliquescence"*

**Response**: Many thanks. We have revised in the following sentence and now they read as:

**Page 4 line 85-87:** "To accurately measure phase transition (e.g., DRH and ERH), a highly stable measurement condition is essential, especially maintaining a small temperature perturbation in the humidification system and inside the second DMA to prevent pre-deliquescence."

*9) Line 85: Remove "the"*

**Response:** Thank you for your comment. I cannot find "the" in line 85, but I checked "the" in the whole of manuscript. All revision is marked with red in the manuscript.

*10) Line 98: Change to "we present the design of a nano-HTDMA. . . ."*

**Response:** Many thanks. We have revised in the following sentence and now they read as:

**Page 4 line 98-99:** "In this study, we present the design of a nano-HTDMA setup that enables high accuracy and precision in hygroscopic growth measurements of aerosol nanoparticles with diameters less than 10 nm."

*11) Line 109: Change to "We designed a nano-HTDMA system to. . . ."*

**Response:** Many thanks. We have revised in the following sentence and now they read as:

**Page 5 line 109-110:** "We designed a nano-HTDMA system to measure the aerosol nanoparticle hygroscopic growth factor ($g_f$), especially aiming for accurate measurement of phase transition and hygroscopic growth factor for nanoparticles in the sub-10 nm size range."

*12) Lines 126, 128, 130, 145: Please don't use abbreviations "Di." and "L.", and please use metric units instead of inches (").*

**Response:** Many thanks. We have revised in the following sentences and now they read as:

**Page 6 line 125-130:** "In the deliquescence mode, dry nanoparticles are humidified by a Nafion humidifier (NH-1, TROPOS Model ND.070, Length 60 cm) to a target RH. In the efflorescence mode, nanoparticles are first exposed to a high RH condition (~97% RH) in a Nafion humidifier (NH-2, Perma Pure Model MH-110, Length 30 cm) and then dried to a target RH through NH-1. The humid flow in the outer tube of NH-1 is a mixture of high-humidity air produced with a custom-built Gore-Tex humidifier and heater (GTHH: TROPOS Model, Inner Radius 1.5 cm & Length 30 cm) and dry air in variable proportions."

**Page 6 line 145-147:** "In addition, we have tested a longer NH-2 (Perma Pure Model MH-110, Length 121 cm) in the efflorescence mode, and no significant difference in measured growth factors are found, indicating that the residence time in NH-1 and NH-2 should be sufficient."

*13) Line 134: Missing a space between "NH-1" and "for"*

**Response**: Many thanks. We have revised in the following sentence and now they read as:

**Page 6 line 134:** "The residence time is ~5.4 s in the NH-1 for both the deliquescence and the efflorescence modes."

14) Change from "Pure" to "Perma Pure"

**Response:** Many thanks. We have revised in the following sentence and now they read as:

**Page 7 line 148-151:** "The number size distribution of the humidified nanoparticles is measured with a combination of the second nano-DMA (nano-DMA2) and the ultrafine CPC. Similar to Biskos et al. (2016b), a multiple Nafion humidifier (NH-3, Perma Pure Model PD-100) is used in our nano-HTDMA system to rapidly adjust the RH of the sheath flow of nano-DMA2."

*15) Line 161: Change to "a low flow resistance. . . ."*

**Response:** Many thanks. We have revised in the following sentence and now they read as:

**Page 7 line 161-162:** "In order to minimize the pressure drop along the recirculating sheath flow loop, a low flow resistance MFM and hydrophobic filter (HF: Whatman Model 6702-3600) are used."

*16) Line 218: Change from "Thence" to "Thus"*

**Response:** Many thanks. We have revised in the following sentence and now they read as:

**Page 9-10 line 218-219:** "Thus, accurate calibrations of sheath flow rates and HV are crucial for constraining the uncertainty associated with sizing of nanoparticles below 100 nm."

*17) Line 258: Change from "high" to "highly"*

**Response:** Many thanks. We have revised in the following sentence and now they read as:

**Page 11 line 258-262:** "Since all our temperature sensors and the highly accurate DPM (EDGE TECH Model MIRROR-99) are installed in the aforementioned well-insulated chamber and the chamber temperature is maintained with air conditioner at about 292.15±0.1 K, we calibrate the temperature sensors and correct their systematic shift by comparing the record of temperature sensors and the DPM by keeping them in parallel inside the chamber over a 12-hour time period."

*18) Line 273: Change from "atomizing" to "atomized from a . . . ."*

**Response:** Many thanks. We have revised in the following sentence and now they read as:

**Page 12 line 272-274:** "Also, 100-nm PSL nanoparticles were atomized from a PSL solution of mixing 3 drops of 100-nm PSL with 300 mL distilled and de-ionized milli-Q water."

*19) Line 298: Change from "select" to "to select"*

**Response:** Many thanks. We have revised in the following sentence and now they read as:**Page 13 line 298-300:** "Afterwards, when using nano-DMA1 to select 100 nm PSL, the scanned size

distribution by nano-DMA2 has a peak diameter at 100.3 nm (Fig. 3b), indicating a good sizing accuracy of the nano-DMA1 too."

*20) Line 302: Change from "to estimate" to "estimating"*

**Response:** Many thanks. We have revised in the following sentence and now they read as:

**Page 13 line 301-303:** "Duplissy et al. (2009) and Wiedensohler et al. (2012) suggested estimating the sizing accuracy of sub-100 nm nanoparticles through the DMA transfer function."

*21) Line 303: Place a "the" in front of "DMA"*

**Response:** Many thanks. We have revised in the following sentence and now they read as:

**Page 13 line 301-303:** "Duplissy et al. (2009) and Wiedensohler et al. (2012) suggested estimating the sizing accuracy of sub-100 nm nanoparticles through the DMA transfer function."

*22) Line 304: "HV" is already defined.*

**Response:** Many thanks. We have revised in the following sentence and now they read as:

**Page 13 line 303-305:** "The theoretical DMA transfer function (see SI. S2. Eq. (S2-S4)) was proposed by Knutson and Whitby (1975) and they noted that sizing is crucially dependent on flow rates and HV applied to the DMA."

**Page 9-10 line 218-219:** "Thus, accurate calibrations of sheath flow rates and HV are crucial for constraining the uncertainty associated with sizing of nanoparticles below 100 nm."

23) Line 323: Change "to calibrate" to "the calibration of"

**Response:** Many thanks. We have revised in the following sentence and now they read as:

**Page 14 line 323-324:** "Note that, we also tested the calibration of the DMA voltage with a voltage meter with lower accuracy of ±1%, and the DMA voltages can only be kept within ±1% around the set value."

*24) Line 343: Change "deliquesce" to "deliquescence"*

**Response:** Many thanks. We have revised in the following sentence and now they read as:

**Page 15 line 341-344:** "Since our RH sensors were all well calibrated and the uncertainty of RH measurement is ±1%, it is reasonable to hypothesize that the RH upstream of nano-DMA2 has already reached the deliquescence RH of ammonium sulfate nanoparticles."

*25) Line 384: Change "undergo" to "undergone"*

**Response:** Many thanks. We have revised in the following sentence and now they read as:

**Page 16 line 381-384:** "Biskos et al. (2006b, 2007) attributed these two modes to the co-existence of solid and liquid phase nanoparticles at RH close to the DRH of ammonium sulfate, due to the slight inhomogeneity of RH in the second nano-DMA, i.e., some nanoparticles have already undergone deliquescence (liquid state) and some are not (solid)."

*26) Line 389: Change to "As RH further increases, the peak. . . ."*

**Response:** Many thanks. We have revised in the following sentence and now they read as:

**Page 17 line 389-391:** "As RH further increases, the peak diameter of normalized number size distribution of the blue mode increases, indicating the continuous growth the nanoparticles after deliquescence."

*27) Line 445: Change to "in the hygroscopic growth factor. . . . "*

**Response:** Many thanks. We have revised in the following sentence and now they read as:

**Page 19 line 444-446:** "For example, a slight increase in the hygroscopic growth factor of 6-nm ammonium sulfate nanoparticles is observed in the RH range from 65 to 79% RH before deliquescence."

*28) Line 454: Change "a" to "an"*

**Response:** Many thanks. We have revised in the following sentence and now they read as:

**Page 19 line 450-454:** "Note that, the ammonium sulfate hygroscopic data from Biskos et al. (2006b) shown here are all generated by an electrospray, but in our experiments, only the ammonium sulfate nanoparticles with diameters smaller than 20 nm (i.e., 10, 8, and 6 nm) were generated by an electrospray, while the larger nanoparticles (i.e., 20, 60, and 100 nm) were generated by an atomizer."

*29) Lines 460, 461: Remove hyphen before "nm"*

**Response:** Many thanks. We have revised in the following sentence and now they read as:

**Page 20 line 461-464:** "Figure S12a shows a ~ 0.1 higher growth factor of 20 nm ammonium sulfate generated by the electrospray than that using the atomizer in the RH range from 55% to 82%, which is similar to the difference in hygroscopic growth factor of 20 nm NaCl aerosol nanoparticles using the different generation method as observed in Fig S12b in Biskos et al. (2006a)."

*30) Line 472: Change to "the hygroscopicity of. . . ."*

**Response:** Many thanks. We have revised in the following sentence and now they read as:

**Page 20 line 471-473:** "As a common constituent of atmospheric aerosol particles (Tang and Munkelwitz, 1993, 1994; Tang 1996; Tang et al., 2007), the hygroscopicity of sodium sulfate with diameters above 20 nm particles has been investigated by a few groups (Tang et al., 2007; Xu and Schweiger, 1999; Hu et al., 2010)."

*31) Line 482: Change to "an external mixture of . . . ."*

**Response:** Many thanks. We have revised in the following sentence and now they read as:

**Page 20-21 line 479-482:** "Two intersecting modes in the measured number size distribution of humidified sodium sulfate nanoparticles are observed at RH close to the DRH (Fig. S9 and S10 in the Supplementary Information) and ERH, suggesting an external mixture of aqueous and solid nanoparticles."

*32) Line 488: Change to "RH=84%"*

**Response:** Many thanks. We have revised in the following sentence and now they read as:

**Page 21 line 488-492:** "For example, at RH=84%, the hygroscopic growth factor of 6 nm sodium sulfate is only ~ 1.3 (in efflorescence mode), while the respective growth factors are about 1.5 and 1.8 for 20 nm and 14-16 μm particles."

*33) Line 515: Change "underline" to "underlying".*

**Response:** Many thanks. We have revised in the following sentence and now they read as:

**Page 22 line 513-516:** "As different hydrates of sodium sulfate may exist during the deliquescence and efflorescence processes (Xu and Schweiger, 1999), to explain the underlying mechanism of the size dependent hygroscopicity of sodium sulfate particles can be challenging."